# Physiological and Transcriptional Responses of *Apocynum venetum* to Salt Stress at the Seed Germination Stage

**DOI:** 10.3390/ijms24043623

**Published:** 2023-02-11

**Authors:** Xin Li, Jinjuan Li, Hongyan Su, Ping Sun, Zhen Zhang, Mengfei Li, Hua Xing

**Affiliations:** 1State Key Laboratory of Aridland Crop Science, College of Agronomy, Gansu Agricultural University, Lanzhou 730070, China; 2Institute of Livestock Grass and Green Agriculture, Gansu Academy of Agricultural Sciences, Lanzhou 730070, China; 3College of Life Science and Technology, Gansu Agricultural University, Lanzhou 730070, China

**Keywords:** *Apocynum venetum*, salt stress, seed germination, physiological change, transcriptomic analysis, gene expression

## Abstract

*Apocynum venetum* is a semi-shrubby perennial herb that not only prevents saline–alkaline land degradation but also produces leaves for medicinal uses. Although physiological changes during the seed germination of *A. venetum* in response to salt stress have been studied, the adaptive mechanism to salt conditions is still limited. Here, the physiological and transcriptional changes during seed germination under different NaCl treatments (0–300 mmol/L) were examined. The results showed that the seed germination rate was promoted at low NaCl concentrations (0–50 mmol/L) and inhibited with increased concentrations (100–300 mmol/L); the activity of antioxidant enzymes exhibited a significant increase from 0 (CK) to 150 mmol/L NaCl and a significant decrease from 150 to 300 mmol/L; and the content of osmolytes exhibited a significant increase with increased concentrations, while the protein content peaked at 100 mmol/L NaCl and then significantly decreased. A total of 1967 differentially expressed genes (DEGs) were generated during seed germination at 300 mmol/L NaCl versus (vs.) CK, with 1487 characterized genes (1293 up-regulated, UR; 194 down-regulated, DR) classified into 11 categories, including salt stress (29), stress response (146), primary metabolism (287), cell morphogenesis (156), transcription factor (TFs, 62), bio-signaling (173), transport (144), photosynthesis and energy (125), secondary metabolism (58), polynucleotide metabolism (21), and translation (286). The relative expression levels (RELs) of selected genes directly involved in salt stress and seed germination were observed to be consistent with the changes in antioxidant enzyme activities and osmolyte contents. These findings will provide useful references to improve seed germination and reveal the adaptive mechanism of *A. venetum* to saline–alkaline soils.

## 1. Introduction

*Apocynum venetum* L. (Apocynaceae) is a semi-shrubby perennial herb that is widely distributed in saline–alkaline soils and plays a critical role in preventing the land from degradation [1,2]. The leaves are used as a traditional Chinese medicine (TCM) for the treatment of cardiac disease, hypertension, and nephritis, which largely rely on the bioactive compounds such as flavonoids, organic acids, and polysaccharides [3,4,5].

In natural habitats, the plants have to face abiotic stresses, such as drought, extreme temperatures, and high salinity [6]. In saline–alkaline soils, the plants grow flourishingly due to their adaptive ability in a high salinity environment [7,8]. Seed germination is the initial stage throughout the life cycle and plays a determining role in the distribution of plant populations, especially under adverse abiotic conditions [9,10]. Currently, germination characteristics and physiological changes have been conducted during the seed germination of *A. venetum* in response to abiotic stresses. Specifically, salt and drought stress significantly inhibited seed germination [11,12,13,14]; fluctuated temperatures (10/25 and 15/30 °C, 12 h/12 h) were more in favor of seed germination than other temperatures [8]; a prolonged storage period significantly decreased seed germination [8,15]; and there was no significant effect of light on seed germination [8]. For the physiological changes in response to salt stress, the rate of electrolyte leakage, the content of osmolytes (e.g., soluble sugar, malondialdehyde (MDA), and proline (Pro)), and the activity of antioxidant enzymes (e.g., superoxide dismutase (SOD), ascorbateperoxidase (APX), catalase (CAT), and peroxidase (POD)) gradually increased with increased NaCl concentrations [13,16,17,18]. Meanwhile, salt stress induced the UR of genes involved in cation transport and antioxidants while inducing the DR of genes involved in cell wall structure [19].

Previous studies have found that there are distinct resistances for seed germination among different psammophytes under salt stress. For example, the seed germination of *Halimodendron halodendron* and *Alhagi sparsifolia* was significantly affected by the NaCl treatments [20]; the seed germination of *Halogeton glomeratus* and *Cannabis sativa* was inhibited by increased salt concentrations [21,22]. Extensive studies have demonstrated that plants respond and adapt to salt stress through various physiological, biochemical, and transcriptional processes, thereby acquiring stress tolerance [23,24]. For example, the content of soluble protein and Pro increased at an early stage and then decreased with the prolonging of salt stress, and the activity of SOD, POD, and CAT as well as the content of MDA in *Broyssonetia papyrifera* and *Pogostemon cablin* gradually increased with increased salt concentrations [25,26]. Hundreds of genes involved in abiotic tolerance as well as growth and development have also been identified in response to salt stress [27]. For example, the stress-responsive genes (e.g., DREB/CBFs), abiotic stress-related TFs (e.g., ZAT, WRKY, and NAC), and transporters (e.g., MATE and LeOPT1-like) were up-regulated in *Arabidopsis* under salt stress [28,29,30]. The above results indicate that salt stress triggers a wide range of responses, including physiological characteristics (e.g., seed germination and growth), cellular metabolism (e.g., osmolytes accumulation and enzyme activity), and molecular regulation (e.g., gene and protein expression).

To date, a low-salt environment improving the seed germination of *A. venetum* has been found, and the physiological changes have also been primarily conducted [16,17,18], while the adaptive mechanism to salt stress is still limited. In order to reveal the adaptive mechanism of *A. venetum* to salt stress in this study, we examined the changes in the rate of seed germination, the activity of antioxidant enzymes, the content of osmolytes, and the expression levels of genes in *A. venetum* at the seed germination stage under different NaCl treatments.

## 2. Results

### 2.1. Seed Germination Rate at Different NaCl Treatments

As shown in Figure 1, the seed germination rate was distinctly affected by different NaCl treatments (0–300 mmol/L, *p* < 0.05), with an increase at the low NaCl concentration (50 mmol/L), while there was a gradual decrease with increased NaCl concentrations (from 100 to 300 mmol/L) compared with CK.

### 2.2. Antioxidant Enzyme Activities at Different NaCl Treatments

As shown in Figure 2, significant changes in the activities of the four antioxidant enzymes (i.e., SOD, POD, CAT, and APX) in seeds were observed under different NaCl treatments, with a significant increase with NaCl concentrations from 0 (CK) to 150 mmol/L and a significant decrease with NaCl concentrations from 150 to 300 mmol/L.

### 2.3. Osmolyte Contents under Different NaCl Treatments

As shown in Figure 3, significant changes in osmolyte contents (i.e., soluble sugar, protein, MDA, and Pro) in seeds were observed under different NaCl treatments, with a significant increase for the contents of soluble sugar, MDA, and Pro with increased NaCl concentrations, while a significant decrease in the protein content was observed at NaCl concentrations from 100 to 300 mmol/L.

### 2.4. Transcriptomic Analysis

#### 2.4.1. Global Gene Analysis

In order to find far more genes involved in salt tolerance, transcriptomic analysis was conducted on a higher salt concentration (300 mmol/L) instead of a lower salt concentration (50 mmol/L) vs. CK. After data filtering, 51.43 and 65.36 million high-quality reads were generated, and 42.12 and 53.26 million unique reads with 0.81 and 0.80 million multiple reads were mapped for CK and NaCl (300 mmol/L), respectively (Table 1; Appendix A).

A total of 52,305 unigenes were annotated on the Kyoto Encyclopedia of Genes and Genomes (KEGG, 34,198), Eukaryotic Orthologous Groups of proteins (KOG, 22,630), NCBI non-redundant protein (Nr, 36,077), and SwissProt (27,467) databases (Appendix A). Using the KEGG database, 1967 DEGs were enriched into 139 metabolism pathways such as global and overview maps, carbohydrate metabolism, and amino acid metabolism (Appendix A). Using the KOG database, 43.27% unigenes encoded the identified proteins that could be classified into 25 functional categories (Appendix A). Using the NR database, the top 10 species included: *Coffea arabica*, *Carica papaya*, *Coffea eugenioides*, *Pistacia vera*, *Actinidia chinensis*, *Vitis vinifera*, *Citrus clementine*, *Theobroma cacao*, *Olea europaea*, and *Citrus sinensis* (Appendix A). Using the SwissProt database, 52.51% unigenes were annotated to have biological functions. Using the Gene Ontology (GO) database, the DEGs were classified into three ontologies, including biological process (BP), cellular component (CC), and molecular function (MF) (Appendix A).

#### 2.4.2. Identification of DEGs

Among the 1967 DEGs, 1626 genes were UR and 341 genes were DR at 300 mmol/L NaCl vs. CK (Figure 4A,B), based on the principal component analysis (Appendix A) and Pearson correlation analysis (Appendix A). The cluster heat map of the 1967 DEGs is shown in Figure 4C. The distinct difference in the expression levels of genes between NaCl and CK treatments indicates that the data can be used for further analysis of gene biological functions.

#### 2.4.3. Distribution and Classification of DEGs

Of these 1967 DEGs, 1729 genes were identified from the KEGG, KOG, SwissProt, or GO databases (Figure 5A). Among the 1729 genes, 1487 genes with characterized biological functions were distributed into 1293 UR and 194 DR (Figure 5B). Based on their biological functions, the 1487 genes were classified into 11 categories, including salt stress (29), stress response (146), primary metabolism (287), cell morphogenesis (156), transcription factor (TFs, 62), bio-signaling (173), transport (144), photosynthesis and energy (125), secondary metabolism (58), polynucleotide metabolism (21), and translation (286) (Figure 5C).

### 2.5. Specific Classification of DEGs and Validation of Expression Levels

#### 2.5.1. DEGs Directly Associated with Salt Stress

Based on the biological functions of proteins encoded by the DEGs, 29 genes (25 UR, 4 DR) were found to be directly associated with salt stress (e.g., *ALDH7B4*, *ALDH10A8*, and *ANN2*) (Table 2). The expression levels of 10 selected genes were validated by qRT-PCR, with 3.07- (*At2g40140*) to 18.54-fold (*IQM4*) UR for eight genes and 0.47- (*PCC13*) and 0.50-fold (*WSD1*) DR for two genes at the 300 mmol/L NaCl vs. CK (Figure 6). Meanwhile, the RELs were consistent with their Reads Per kb per Million (RPKM) values (Table 2).

#### 2.5.2. DEGs Directly Associated with Antioxidant Enzymes

In this study, 23 genes (21 UR, 2 DR) directly associated with antioxidant enzymes, including SOD (*SODCP*, *FSD2,* and *SODA*), POD (e.g., *At5g06290*, *PNC1,* and *APX3*), and CAT (*CATA, CAT1,* and *PNC1*) (Table 3), were screened from the 146 DEGs involved in stress response, and another 123 DEGs were directly associated with stress responses such as temperature, pathogens, and oxidative stresses (Appendix A). The expression levels of 12 selected genes were validated by qRT-PCR, with 4.81- (*CAT1*) to 18.34-fold (*SODA*) UR for 11 genes and 0.71-fold (*poxN1*) DR at 300 mmol/L NaCl vs. CK (Figure 7). Meanwhile, the RELs were consistent with their RPKM values (Table 3).

#### 2.5.3. DEGs Directly Associated with Soluble Sugar and Protein Metabolism

In this study, 79 genes (72 UR, 7 DR) directly associated with osmolytes, including soluble sugar (41 genes) and protein (38 genes), were screened from the 287 DEGs involved in primary metabolism; specifically, the 41 genes associated with soluble sugar metabolism included glucose (e.g., *G6pc2*, *G6PDC*, and *GAPB*), sucrose (*INVE*, *SUS2,* and *SUS3*), fructose (e.g., *FKFBP*, *RHVI2,* and *FRK2*), galactose (*GOLS1* and *GOLS2*), trehalose (*TPS6* and *TPS7*), fucose (*OFUT19* and *OFUT39*), and starch (*SBEI*, *DPE2,* and *AMY3*), and the 38 genes associated with protein metabolism included *PSMD2*, *RPN1A, RPN10*, etc. (Table 4). The other 208 genes were associated with fatty acids, lipid metabolism, amino acids, etc. (Appendix A). The expression levels of the 23 selected genes were validated by qRT-PCR, with 1.97- (*ALEU*) to 12.13-fold (*GOLS1*) UR for 22 genes and 0.57-fold (*Cys*) DR at 300 mmol/L NaCl vs. CK (Figure 8). Meanwhile, the RELs were consistent with their RPKM values (Table 4).

#### 2.5.4. DEGs Directly Associated with Cell Morphogenesis for Seed Germination

In this study, six genes (5 UR, 1 DR) directly associated with seed germination (i.e., *F16*, *FIP1*, *At5g13200*, *At5g23350*, *KAI2,* and *ROH1*) (Table 5) were screened from the 156 DEGs involved in cell morphogenesis. The other 150 genes were associated with flower development, cell wall organization, programmed cell death, etc. (Appendix A). The expression levels of the six genes were validated by qRT-PCR, with 1.42- (*At5g23350*) to 14.48-fold (*F16*) UR at 300 mmol/L NaCl vs. CK (Figure 9). Meanwhile, the RELs were almost consistent with their RPKM values, except for the *At5g23350* gene (Table 5).

#### 2.5.5. TFs Directly Associated with Stress Response and Seed Germination

In this study, 16 TFs (15 UR, 1 DR) directly associated with stress response and seed germination, including *MYB* (*MYB73* and *MYB1R1*), *WRKY* (e.g., *WRKY4*, *WRKY23,* and *WRKY24*), *NAC* (e.g., *NAC083*, *NAC091,* and *NAC100*), *BZIP44*, *TCP14,* and *UNE10* (Table 6), were screened from the 62 TFs; the other 46 TFs were associated with biotic stress, the regulation of transcription by RNA polymerase II, flower development, etc. (Appendix A). The expression levels of eight selected genes were validated by qRT-PCR, with 4.33- (*JA2L*) to 12.13-fold (*MYB73*) UR at 300 mmol/L NaCl vs. CK (Figure 10). Meanwhile, the RELs were consistent with their RPKM values (Table 6).

#### 2.5.6. DEGs Directly Associated with Hormone Response

In this study, 36 genes (30 UR, 6 DR) directly associated with hormone response, including GA (*GASA2*), IAA (e.g., *ABP19A*, *AUX22D*, and *AUX12KD*), CTK (*CISZOG2*, *AHK3*, and *ARR4*), JA (*JOX2* and *TIFY6B*), ABA (e.g., *PYL3*, *PYL4,* and *GRDP1*), and ETH (e.g., *RAV1*, *ETR2*, and *ERF4*) response (Table 7), were screened from the 173 DEGs involved in bio-signaling. The other 137 genes were associated with other bio-signals such as protein kinases, protein phosphatase, calcium sensors, etc. (Appendix A). The expression levels of the 12 selected genes were validated by qRT-PCR, with 3.92- (*JOX2*) to 13.36-fold (*PP2CA*) UR for eight genes and 0.39- (*ETR2*) to 0.98-fold (*ERF013*) DR for four genes at 300 mmol/L NaCl vs. CK (Figure 11). Meanwhile, the RELs were almost consistent with their RPKM values (Table 7).

#### 2.5.7. DEGs Directly Associated with Ion Transport

In this study, 34 genes (27 UR, 7 DR) directly associated with ion transport (e.g., *AHA10*, *CSC1,* and *CLC-B*) (Table 8) were screened from the 144 DEGs involved in transport, and the other 110 genes were associated with other kinds of transport such as sugar transport, protein transport, amino acid transport, etc. (Appendix A). The expression levels of the 10 selected genes were validated by qRT-PCR, with 4.07- (*CAX1*) to 12.41-fold (*KEA2*) UR for nine genes and 0.78-fold (*ZIP1*) DR at 300 mmol/L NaCl vs. CK (Figure 12). Meanwhile, the RELs were consistent with their RPKM values (Table 8).

#### 2.5.8. DEGs Associated with Other Biological Functions

There are 490 DEGs involved in other biological functions, including photosynthesis and energy (125 genes, Appendix A), secondary metabolism (58 genes, Appendix A), polynucleotide metabolism (21 genes, Appendix A), and translation (286 genes, Appendix A). These genes may also participate in the seed germination of *A. venetum* in response to salt stress.

## 3. Discussion

Under both natural and agricultural conditions, plants are frequently exposed to unfavorable environments such as salt stress, extreme temperatures, and drought stress, which restrict plant distribution and reproduction, while plant adaptations that are genetically determined confer stress resistance [31]. Salt stress is a major abiotic stress that hinders seed germination, growth and development, and flowering and fruiting by driving various physiological and biochemical changes such as nutrition uptake, photosynthesis, and cellular metabolism [32,33,34,35,36,37]. In this study, we found that salt stress significantly affected the seed germination of *A. venetum* by changing the activity of antioxidant enzymes (i.e., SOD, POD, CAT, and APX), the content of osmolytes (i.e., soluble sugar, protein, MDA, and Pro), and the expression levels of genes associated with salt-stress response and seed germination.

Extensive experiments have demonstrated that stress resistance in response to salt stress is involved in the changes in physiological parameters (e.g., seed germination rate, plant growth, and flower development), antioxidant enzyme activities (e.g., SOD, POD, and CAT), and osmolyte contents (e.g., soluble protein, MDA, and Pro) [23,38]. For example, the seed germination of *A. venetum* decreases sharply under salt stress [11,12,13,14]. Salt stress causes oxidative stress that is responsible for the generation of reactive oxygen species (ROS), which are highly reactive and cause damage to biomolecules (e.g., lipids, proteins, and nucleic acids). To minimize the effect of oxidative stress, the ROS generation and increased activity of many antioxidant enzymes (e.g., SOD, CAT, and APX) during salt stress can protect plants from damage [23,39,40,41,42,43,44]. Under salt stress, ion imbalance and water deficiency induce osmotic stress, which triggers osmotic signaling pathways that regulate related genes’ expression, the enzyme activity of osmolyte biosynthesis, and the contents of osmolytes (e.g., soluble sugar, MDA, and Pro) [17,24,45].

In this study, we found that a low NaCl concentration (50 mmol/L) could promote the seed germination of *A. venetum* by adjusting the antioxidant enzyme activities and osmolyte contents. Actually, salt response is not only involved in the changes in phenotypes, antioxidant enzymes, and osmolytes but also in transcriptional alternations [46,47]. Here, 1487 DEGs (1293 UR and 194 DR) were observed during the seed germination of *A. venetum* at 300 mmol/L NaCl vs. CK, with 11 categories classified, including response to salt, stress response, primary metabolism, cell morphogenesis, TFs, bio-signaling, transport, photosynthesis and energy, secondary metabolism, polynucleotide metabolism, and translation.

Specifically, a total of 29 genes were directly associated with salt stress. For example, *ALDH7B4* is activated by high salinity, dehydration, and ABA in a tissue-specific manner [48]; annexin (*ANNs*) in *Arabidopsis* is regulated by exposure to salt, drought, and extreme temperature conditions [49]; and *RD22* is induced by salt stress and water deficit during the early and middle stages of seed development [50]. *CAMBP25* is induced in *Arabidopsis* seedlings exposed to high salinity, dehydration, and low temperature [51]. *PCC13* is abundantly expressed in dried leaves and abscisic acid-treated dried callus and is involved in the response to salt stress [52]; *RBG7* plays a role in altering the germination and seedling growth of *Arabidopsis* under various stresses [53]; *IQM4* plays a key role in modulating the responses to salt, ABA, and osmotic stress during seed germination and post-germination growth [54]; *NFD4* mediates the transport of Na^+^ and K^+^ into the vacuole, influences plant development, and contributes to salt tolerance [55]; *WSD1* is involved in the accumulation of wax esters in response to salt, drought, and ABA [56]; and *At2g40140* plays important roles in modulating the tolerance of *Arabidopsis* to salt stress [57]. In this study, most of the genes involved in salt stress were UR, which may play a determining role in obtaining the ability of seed germination of *A. venetum* in saline–alkaline soils.

The primary effect of salt stress is hyperionic and hyperosmotic stresses, as well as, in severe cases, oxidative stress in plants. Oxidative stress is responsible for the generation of ROS, which are deleterious to plants. To minimize the effect of oxidative stress, plants have developed an efficient system of antioxidant enzymes (e.g., SOD, CAT, and APX) that can protect plants from damage [39,42,43,58,59,60,61]. Here, 23 genes were directly associated with antioxidant enzyme activities. For example, the SOD (SODCP, FSD2, and SODA) is a critical enzyme responsible for the elimination of superoxide radicals and is considered to be a key antioxidant in aerobic cells [23]; the POD (At5g06290, APX3, poxN1, PRDX1, PEX11C, and GPX1) catalyzes the reduction of hydrogen peroxide and plays a role in cell protection against oxidative stress by detoxifying peroxides [62]; and the CAT (CATA, CAT1, and PNC1) protects cells from the toxic effects of hydrogen peroxide [63]. In this study, most of the genes encoding SOD, POD, and CAT were UR, which is in accordance with the increased activities of SOD, POD, and CAT, which can reduce oxidative stress under salt stress.

Saline and alkaline ions can reduce the soil water potential, which makes it difficult for plants to absorb water, and then causes osmotic stress [35]. Previous studies have demonstrated that genes related to soluble sugar and protein metabolism play critical roles in response to salt stress in other plants. For example, the trehalose-6-phosphate synthase (*TPS*) gene plays a critical role in *Citrullus lanatus*’ response to salt stress [64]; the *OsGolS1* gene is significantly up-regulated in *Oryza sativa* ssp. *japonica* under salt stress [65]; and the overexpression of *MsTRX* in tobacco induced the upregulation of beta-amylase 1 (*BAM1*) under salt stress [66]. In this study, 79 genes were directly associated with soluble sugar and protein metabolism, and 41 genes participate in soluble sugar metabolism. For example, *G6PDC* is involved in the synthesis of D-ribulose 5-phosphate from D-glucose 6-phosphate [67]; *GAPA1* is involved in the synthesis of pyruvate from D-glyceraldehyde 3-phosphate [68]; *PGMP* is involved in both the breakdown and synthesis of glucose [69]; *UGPA* is involved in the UDP-glucose metabolic process [69]; *INVE* participates in cleaving sucrose into glucose and fructose and is associated with the assimilation of nitrogen to control the sucrose to hexose ratio [70]; *SUS2* is involved in providing UDP-glucose and fructose for various metabolic pathways [71]; *FKFBP* is involved in the synthesis and degradation of fructose 2,6-bisphosphate [72]; *F16P2* and *FBA3* are involved in the fructose 1,6-bisphosphate metabolic process [69]; *GOLS1* is involved in the synthesis of raffinose family oligosaccharides and promotes plant stress tolerance [73]; *TPS6* is involved in the trehalose biosynthetic process [74]; *OFUT39* is involved in the fucose metabolic process [75]; *DPE2* is involved in the starch catabolic process and an intermediate on the pathway by which starch is converted to sucrose [76]; *AMY3* is involved in stress-induced starch degradation [77]; *BAM1* plays a role in the starch degradation and maltose metabolism [78]; and *GWD3* is involved in starch degradation and mediates the incorporation of phosphate into starch-like phospho-alpha-glucan [79]. The other 38 genes participate in protein metabolism. For example, *RPN1A* is involved in the ATP-dependent degradation of ubiquitinated proteins [80]; *PCS1* is involved in proteolysis and possesses peptidase activity toward casein [69,81]; *Cys* is involved in the protein catabolic process and shows the highest affinity for Pro [82]; *DEGP1* is involved in the degradation of damaged proteins [83,84]; *PBE2* is involved in the proteasomal protein catabolic process and shows an ATP-dependent proteolytic activity [69]; and *SBT2.5* and *ALEU* are involved in proteolysis and show serine-type peptidase activity [69,85]. In this study, most of the genes involved in soluble sugar and protein metabolism were UR, which is in accordance with the increased content of soluble sugar and protein during seed germination.

For the biological functions of the six genes directly associated with seed germination, *F16* is involved in seed trichome differentiation and expressed in the early phase of cotton fiber development [86]; GEM-like proteins (i.e., FIP1, At5g13200, and At5g23350) are involved in seed germination and play a novel role in regulating the reproductive development of plants [87]; *KAI2* is involved in seed germination and seedling development [88]; and *ROH1* is involved in seed coat development [89]. Thus, the differential expression of these genes could play important roles in seed germination under salt stress.

TFs are emerged as key regulators in various signaling networks and play significant roles by improving the growth and development of plants under stress conditions [90]. Previous studies have demonstrated that genes related to TFs play critical roles in response to salt stress in other plants. For example, the WRKY TFs of cotton play a significant role in the regulation of abiotic stresses (i.e., salt, drought, and extreme temperatures) [91]; MYB TFs are involved in the responses to different abiotic stresses, such as salt, cold, and drought, and the *FvMYB82* gene probably plays an important role in the response to salt and cold stresses in *Arabidopsis thaliana* by regulating downstream related genes [92]; and NAC TFs play vital roles in plant development and responses to various abiotic stresses, and the *ThNAC4* gene of *Tamarix hispida* is involved in salt and osmotic stress tolerance [93]. In this study, 16 TFs were directly associated with stress response and seed germination. Specifically for the stress response, *MYB73* is involved in the salt-stress response [94]; *WRKY TFs* (*WRKY24* and *WRKY71*) play critical regulatory roles in the plant alkaline metal (sodium) salt-stress response [95]; *NAC083* is involved in xylem vessel formation and mediates signaling crosstalk between the salt-stress response and leaf-aging process [96]; and *JA2L* with *MYC2* forms a transcription module that regulates wounding-responsive genes [97]. For seed germination, *BZIP44* is involved in the positive regulation of seed germination through MAN7 gene activation [98]; *TCP14* is involved in the regulation of seed germination and regulates the activation of embryonic growth potential during seed germination [99]; and *UNE10* negatively regulates seed germination [100]. Here, their up-regulation may play a critical role in conferring the seed germination of *A. venetum* under salt stress.

As is known, endogenous hormones play critical roles in plant growth and development. Previous studies have demonstrated that genes related to hormone signaling play critical roles in response to salt stress in other plants. For example, *SmGASA4* was found to be positively regulated by Gibberellin (GA) and significantly enhanced plant resistance to salt, drought, and paclobutrazol (PBZ) stress in *Salvia miltiorrhiza* [101]; Aux/IAA proteins in auxin and ARF transcription factors directly regulate auxin-responsive gene expression and *OsIAA24* and *OsIAA20* are up-regulated in rice under high salt stress [102]; genes related to Cytokinin (CK) and ethylene (ET) were involved in alleviating the root damage of *Tamarix ramosissima* under NaCl stress [103]. In this study, 36 genes were directly associated with hormone response. Specifically, *GASA2* plays a role in the promotion of GA responses during seed germination, flowering, and seed maturation [104]; *ARF2A*, *IAA9*, and *TET4* act as auxin-responsive genes and regulate vegetative growth, lateral root formation, and flower organ senescence [105,106]; *CISZOG2* may regulate active versus storage forms of cytokinin (CTK) and can have an impact on seed growth [107]; *AHK3* acts as a redundant positive regulator of CTK signaling that regulates many developmental processes such as seed germination, cell division, and shoot promotion [108]; *JOX2* prevents the over-accumulation of jasmonate (JA) under stress responses [109]; *TIFY6B* acts as a repressor of JA responses and is involved in the regulation of defense responses [110]; *PYL3* is involved in ABA signaling during seed germination and abiotic stress responses [111]; *PP2CA* acts as a major negative regulator of ABA responses during seed germination and cold acclimation [112]; *ETR2* acts as a redundant negative regulator of ethylene (ETH) signaling [113]; and *ERF013* may be involved in the regulation of gene expression by stress factors and by the components of stress signal transduction pathways [114]. Their differential expression may also play critical roles in promoting the seed germination of *A. venetum*.

Saline and alkaline stress can induce ion toxicity in plants when too many toxic ions enter plant cells, which can harm the plant cytoplasm and organelles; among them, Na^+^ is the main toxicity ion due to the similarity in size of the hydrated ionic radii of Na^+^ and K^+^, which makes them difficult to discriminate [115]. The salt overly sensitive (SOS) regulatory pathway regulates ion homeostasis by modulating Na^+^/H^+^ antiporter activity during salt stress [116]. Previous studies have demonstrated that genes related to ion transport play critical roles in the response to salt stress in other plants. For example, *CLC-c* is involved in the response to salt stress tolerance and seed germination in *Gossypium hirsutum* [117]; K^+^ efflux transporters (*KEAs*) were expressed under abiotic stress (salt, heat, and drought) in *Cajanus cajan* [118]; the Zipper (*Zip*) gene family participates in plant growth and development and the ability to cope with outside environment stresses, which may potentially regulate seed germination and stress resistance in *Miscanthus sinensis* [119]. In this study, 34 genes were directly associated with ion transport. For example, *CSC1* acts as an osmosensitive calcium-permeable cation channel and is activated by hyperosmotic shock after NaCl treatment [120,121]; *CLC-C* is involved in ion transmembrane transport [122]; *RAN1* is involved in copper import into the cell and acts by delivering copper to create functional hormone receptors [123]; *KEA2* transports K^+^ and Cs^+^ preferentially relative to Na^+^ or Li^+^ [124]; *OEP16* is involved in ion transport and acts as a voltage-dependent high-conductance channel [125]; *PPI1* is involved in proton transport and promotes AHA1 plasma membrane ATPase activity by binding to a site different from the 14-3-3 binding site [126]; *NHX2* is involved in the vacuolar ion compartmentalization necessary for cell volume regulation and cytoplasmic Na+ detoxification [127]; *CAX1* translocates Ca^2+^ and other metal ions into vacuoles using the proton gradient formed by H^+^-ATPase and H^+^-pyrophosphatase [128]; *VHA-a2* is involved in vacuolar nutrient storage and in the tolerance to some toxic ions, as well as catalyzing the translocation of protons across membranes [129]; and *ZIP1* mediates zinc uptake from the rhizosphere and may also transport copper and cadmium ions [130]. Their differential expression may also play critical roles in promoting the seed germination of *A. venetum* under salt stress.

Based on the above studies on the physiological and transcriptional changes, a model of the low-salt-stress-enhanced seed germination of *A. venetum* is proposed (Figure 13). Briefly, when seeds are stimulated by salt stress, gene regulation is performed and related genes are differentially expressed; then, bio-signaling such as the hormone response (e.g., *GASA2*, *IAAs*, and *PYLs*) is generated; subsequently, the genes related to salt-stress responses (e.g., *ALDHs*, *ANNs*, and *PCCs*), soluble sugar metabolism (e.g., *G6pc2*, *SUSs*, and *FBAs*), protein metabolism (e.g., *PSMD2*, *PRTs*, and *SBTs*), TFs (e.g., *MYBs*, *WRKYs*, and *NACs*), and ion transport (e.g., *AHA10*, *OEPs*, and *CAXs*) are up-regulated, which promotes the antioxidant enzyme activities (e.g., SOD, POD, and CAT) and osmolyte contents (e.g., soluble sugar, MDA, and Pro) that protect the cells from salt-stress-induced membrane injury; finally, these changes are in favor of building the cell morphogenesis, enhancing seed germination, and conferring the ability to adapt to the saline–alkaline environment.

## 4. Materials and Methods

### 4.1. Plant Materials

Mature seeds of *Apocynum venetum* L. were collected from a resource nursery located at Alxa Left Banner, China (1050 m a.s.l.; E 103°42′14.77″, N 38°18′5.62″) in September 2020. Seeds were air dried at room temperature, stored at 4 °C in air-tight bags in the dark for two months, then cleaned with tap water and successively immersed in 50 °C water for 15 min and sanitized in 70% ethanol for 20 s. After being rinsed with sterile water 5 times, 30 seeds were sown in a Petri dish (15 cm diameter, two layers of gauze on the bottom) containing 10 mL of NaCl solution with 0 (CK), 50, 100, 150, 200, and 300 mmol/L. These doses were selected based on the published literature [11,16,17]. Each treatment had 40 independent biological replicates, which were made in 40 Petri dishes (30 seeds per dish).

### 4.2. Measurement of Germination Rate

After germination in a dark growth chamber at 22 °C for 3 days, the seed germination rate was measured. Three seeds were randomly collected from the 40 independent biological replicates for each NaCl treatment.

### 4.3. Determination of Antioxidant Enzyme Activities

The activities of 4 antioxidant enzymes including SOD, POD, CAT, and APX in seed embryos were determined using a spectrometer (Evolution 201, Thermo Scientific, Waltham, MA, USA). Briefly, SOD activity was determined based on the ability to inhibit the photochemical reduction of nitro blue tetrazolium chloride: extracts (100 μL) were added to the reaction, and the absorbance reading was taken at 560 nm [131]. POD activity was determined by the guaiacol colorimetric method: extracts (1000 μL) were added to the reaction and the absorbance reading was taken at 470 nm [132]. CAT activity was determined by the UV absorption method: extracts (100 μL) were added to the reaction and the absorbance reading was taken at 240 nm [133]. APX activity was determined based on the decrease in the absorbance of the oxidized ascorbate: extracts (100 μL) were added to the reaction and the absorbance reading was taken at 290 nm [134]. Each determination contained three biological replicates (a total of 120 samples = 3 seeds × 40 independent biological replicates) and each biological replicate had three technical replicates.

### 4.4. Determination of Osmolytes Content

The contents of 4 osmolytes including soluble sugar, protein, MDA, and Pro in seed embryos were determined using a spectrometer (Evolution 201, Thermo Scientific, Waltham, MA, USA). Briefly, the soluble sugar content was determined by the phenolsulfuric acid method: extracts (1000 μL) were added to the reaction and the absorbance reading was taken at 485 nm [135]. Protein content was determined by the comassie brilliant blue colorimetric method: extracts (800 μL) were added to the reaction and the absorbance reading was taken at 595 nm [136]. MDA content was determined by the thiobarbituric acid reaction: extracts (7000 μL) were added to the reaction and absorbance readings were taken at 532, 600, and 450 nm [137]. Pro content was determined by the sulfosalicylic acid-acid ninhydrin method: extracts (3000 μL) were added to the reaction and the absorbance reading was taken at 520 nm [138]. Each determination contained three biological replicates (a total of 120 samples = 3 seeds × 40 independent biological replicates) and each biological replicate had three technical replicates.

### 4.5. Transcriptomic Analysis

#### 4.5.1. RNA Extraction, cDNA Library Construction, and Illumina Sequencing

Total RNA samples of CK and 300 mmol/L NaCl treatments with 3×40 biological replicates were extracted using an RNA kit (R6827, Omega Bio-Tek, Norcross, GA, USA). The RNA amount was quantified using a NanoDrop ND1000 spectrophotometer (Nanodrop Technologies, Wilmington, NC, USA), and the quality was determined using an Agilent 2100 Bioanalyzer (Agilent Technologies, Palo Alto, CA, USA). One cDNA library was constructed from the total RNA samples of CK and 300 mmol/L NaCl treatments, and the processes of enrichment, fragmentation, reverse transcription, the synthesis of the second-strand cDNA, and the purification of cDNA fragments were conducted as previously described [139]. Reads were generated using an Illumina HiSeqTM 4000 platform (Gene Denovo Biotechnology Co., Ltd., Guangzhou, China).

#### 4.5.2. Reads Filtration, Assembly, Unigene Expression Analysis, and Basic Annotation

Raw reads were filtered using a FASTQ system to obtain high-quality clean reads by removing reads containing adapters, more than 10% of unknown nucleotides (N), and more than 50% of low-quality (Q-value ≤ 20) bases [140]. Clean reads were assembled using Trinity [141]. The expression level of each transcript was normalized to the RPKM values. The differential expression analysis of transcripts was performed using DESeq2 software [142] between the CK and 300 mmol/L NaCl treatments with the criteria of the false discovery rate (FDR) < 0.05 and |log_2_(fold-change)| > 1. The function of DEGs was annotated using BLAST against the databases, including Nr, KEGG, KOG, SwissProt, and GO, with an e-value ≤ 10^−5^ as a threshold [143].

#### 4.5.3. qRT-PCR Validation

The primer sequences (Appendix A) were designed via a primer blast in NCBI and synthesized by reverse transcription (Sangon Biotech Co., Ltd., Shanghai, China). First, cDNA was synthesized using an RT Kit (KR116, Tiangen, Beijing, China). PCR amplification was performed using a SuperReal PreMix (FP205, Tiangen, Beijing, China). The melting curve was analyzed at 72 °C for 34 s. The *Actin* gene was used as a reference control [14]. The RELs of genes were calculated using a 2*^−^*^ΔΔ*Ct*^ method (*Ct*, cycle threshold value of target gene) according to the following formula [144]. The validation of qRT-PCR contained three biological replicates (a total of 120 samples = 3 seeds × 40 independent biological replicates) and each biological replicate had three technical replicates.
Δ*C_t Test gene_* = *C_t Test gene_* − *C_t Reference gene_*
Δ*C_t Control gene_* = *C_t Control gene_* − *C_t Reference gene_*
−ΔΔ*C_t_* = −(Δ*C_t Test gene_* − Δ*C_t Control gene_*)
Relative gene expression fold (Test gene/Control gene) = 2*^−^*^ΔΔ*Ct*^

### 4.6. Statistical Analysis

SPSS 22.0 software was used for statistical analysis, and one-way analysis of variance (ANOVA) by Duncan’s test was performed for statistical comparisons with *p* < 0.05 considered statistically significant.

## 5. Conclusions

This research reveals that salt stress affects the seed germination of *A. venetum*. The seed germination of *A. venetum* is promoted at low NaCl concentrations such as 50 mmol/L, and significant changes in antioxidant enzyme activities, osmolyte contents, and genes expression levels in *A. venetum* play critical roles in regulating seed germination under different salt stresses. These findings indicate that *A. venetum* plants can adapt to salt stress in saline–alkaline soils by integrating physiological and transcriptional responses. The specific roles of key genes in conferring the ability to salt resistance will require further investigation.

## Figures and Tables

**Figure 1 ijms-24-03623-f001:**
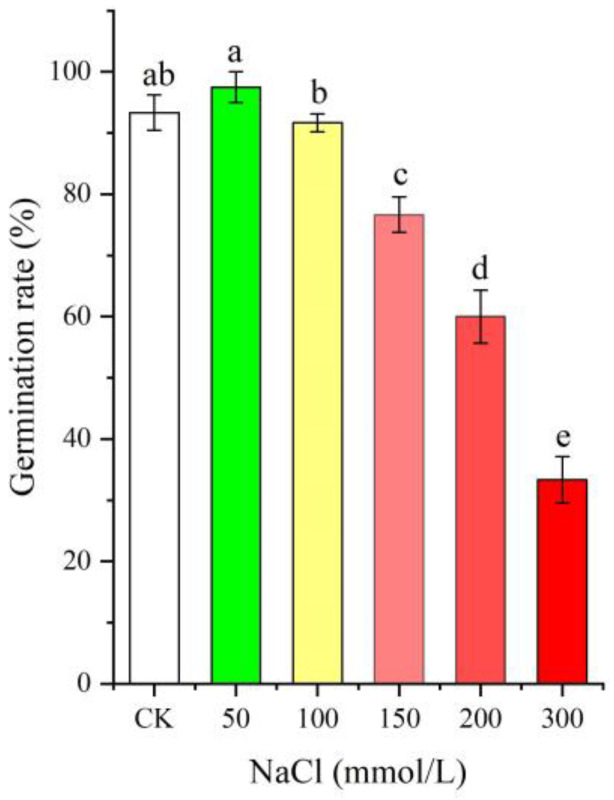
Change in the seed germination rate of *Apocynum venetum* under different NaCl treatments. Different letters represent a significant difference (*p* < 0.05) among different NaCl treatments.

**Figure 2 ijms-24-03623-f002:**
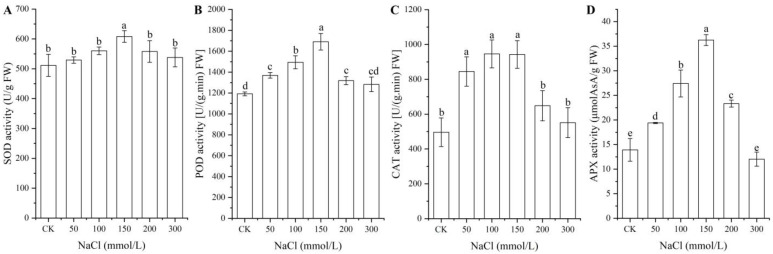
Changes in the activities of the four antioxidant enzymes SOD (**A**), POD (**B**), CAT (**C**), and APX (**D**) in *A. venetum* seeds at different NaCl treatments. Different letters represent a significant difference (*p* < 0.05) among different NaCl treatments.

**Figure 3 ijms-24-03623-f003:**
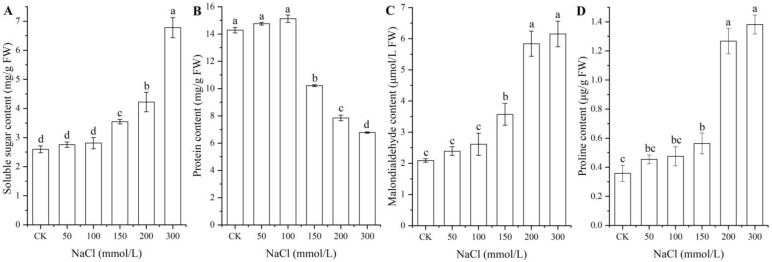
Changes in the contents of soluble sugar (**A**), protein (**B**), MDA (**C**), and proline (**D**) in *A. venetum* seeds under different NaCl treatments. Different letters represent a significant difference (*p* < 0.05) among different NaCl treatments.

**Figure 4 ijms-24-03623-f004:**
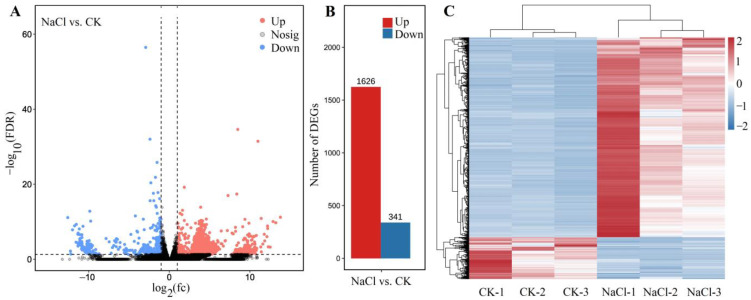
Volcano plot of unigenes (**A**), number of DEGs (**B**), and cluster heat map of the DEGs (**C**) for NaCl (300 mmol/L) vs. CK. The black represent the genes without difference changes in the criteria of the false discovery rate (FDR) < 0.05 and |log2(fold-change)| > 1.

**Figure 5 ijms-24-03623-f005:**
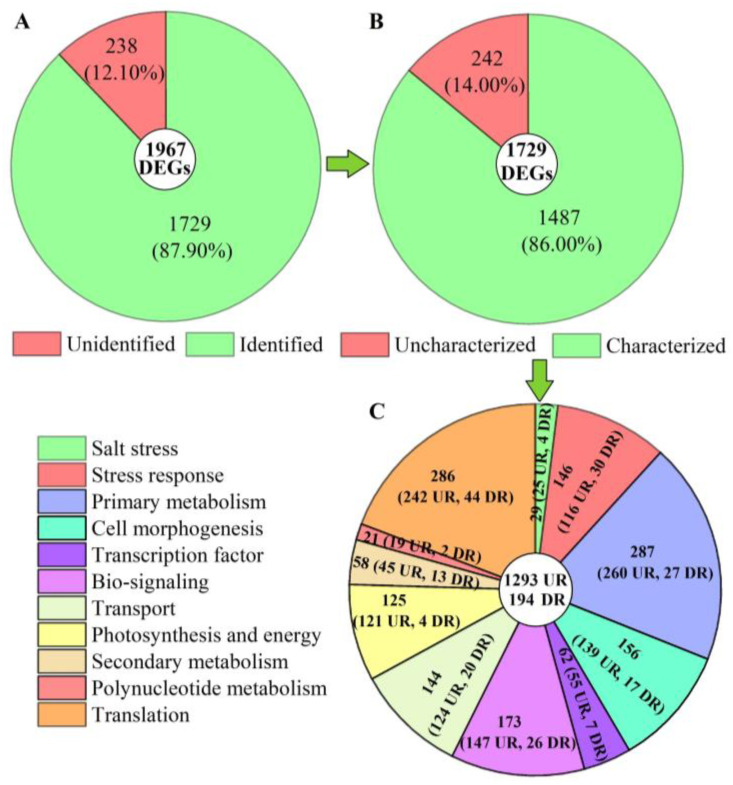
Distribution and classification of DEGs at 300 mmol/L NaCl vs. CK. Abbreviations: UR, up-regulated, DR, down-regulated. Image (**A**) represents the distribution of unidentified and identified genes; image (**B**) represents the distribution of uncharacterized and characterized genes, and image (**C**) represents the classification of the characterized biological functional genes.

**Figure 6 ijms-24-03623-f006:**
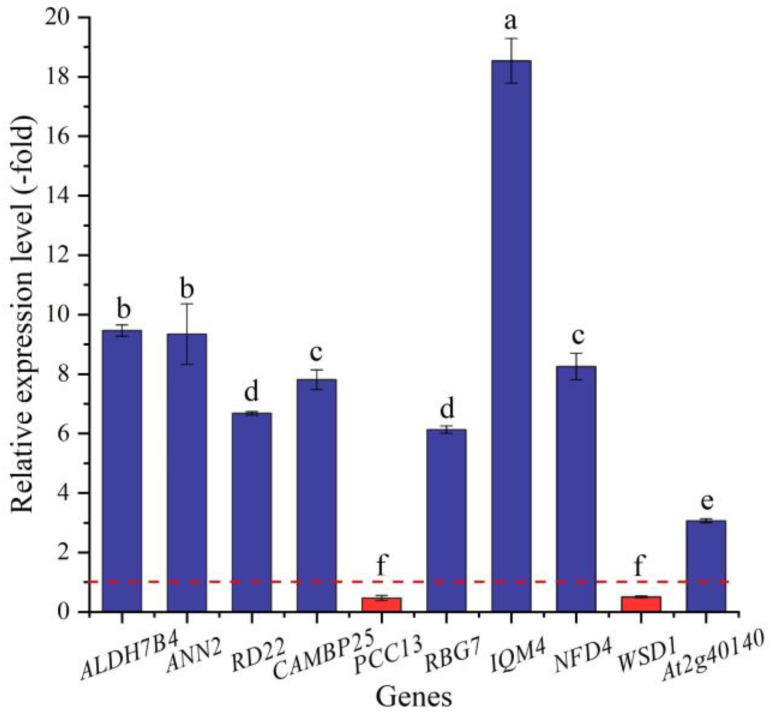
The RELs of genes directly associated with salt stress in *A. venetum* seeds at 300 mmol/L NaCl vs. CK (mean ± SD, *n* = 3). Different letters represent a significant difference (*p* < 0.05) among different genes. The red dotted line differentiates UR (>1) and DR (<1).

**Figure 7 ijms-24-03623-f007:**
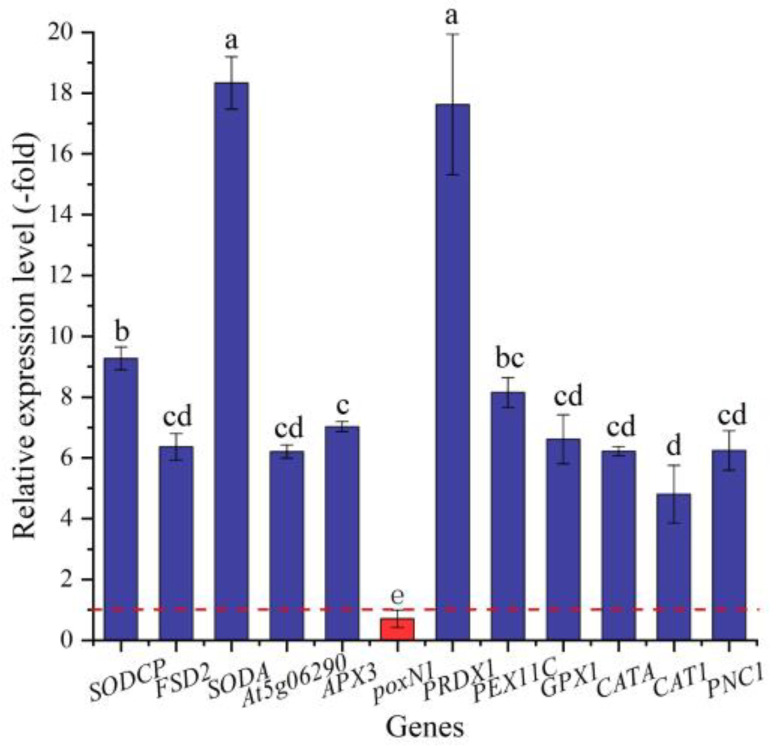
The RELs of genes directly associated with antioxidant enzymes in *A. venetum* seeds at 300 mmol/L NaCl vs. CK (mean ± SD, *n* = 3). Different letters represent a significant difference (*p* < 0.05) among different genes. The red dotted line differentiates UR (>1) and DR (<1).

**Figure 8 ijms-24-03623-f008:**
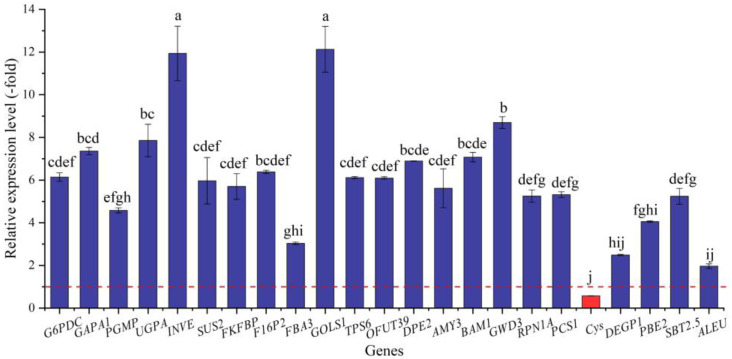
The RELs of genes directly associated with soluble sugar and protein metabolism in *A. venetum* seeds at 300 mmol/L NaCl vs. CK (mean ± SD, *n* = 3). Different letters represent a significant difference (*p* < 0.05) among different genes. The red dotted line differentiates UR (>1) and DR (<1).

**Figure 9 ijms-24-03623-f009:**
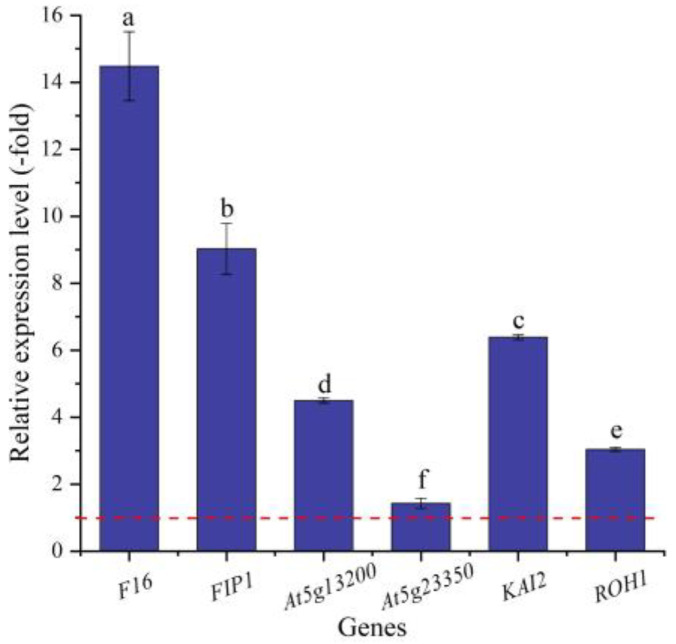
The RELs of genes directly associated with cell morphogenesis for seed germination in *A. venetum* seeds at 300 mmol/L NaCl vs. CK (mean ± SD, *n* = 3). Different letters represent a significant difference (*p* < 0.05) among different genes. The red dotted line differentiates UR (>1) and DR (<1).

**Figure 10 ijms-24-03623-f010:**
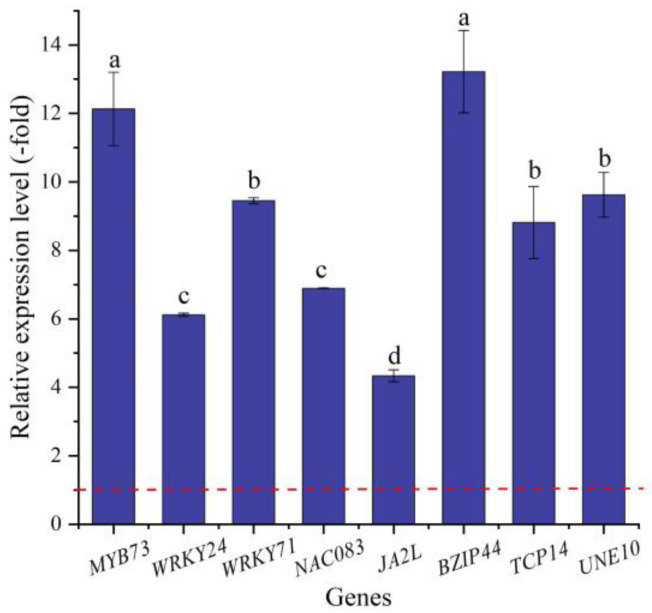
The RELs of TFs directly associated with stress response and seed germination in *A. venetum* seeds at 300 mmol/L NaCl vs. CK (mean ± SD, *n* = 3). Different letters represent a significant difference (*p* < 0.05) among different genes. The red dotted line differentiates UR (>1) and DR (<1).

**Figure 11 ijms-24-03623-f011:**
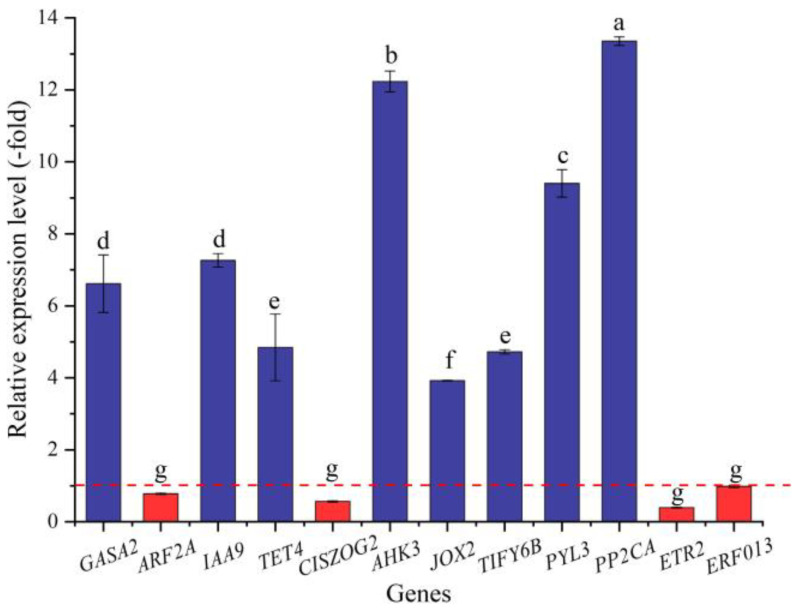
The RELs of genes directly associated with hormone response in *A. venetum* seeds at 300 mmol/L NaCl vs. CK (mean ± SD, *n* = 3). Different letters represent a significant difference (*p* < 0.05) among different genes. The red dotted line differentiates UR (>1) and DR (<1).

**Figure 12 ijms-24-03623-f012:**
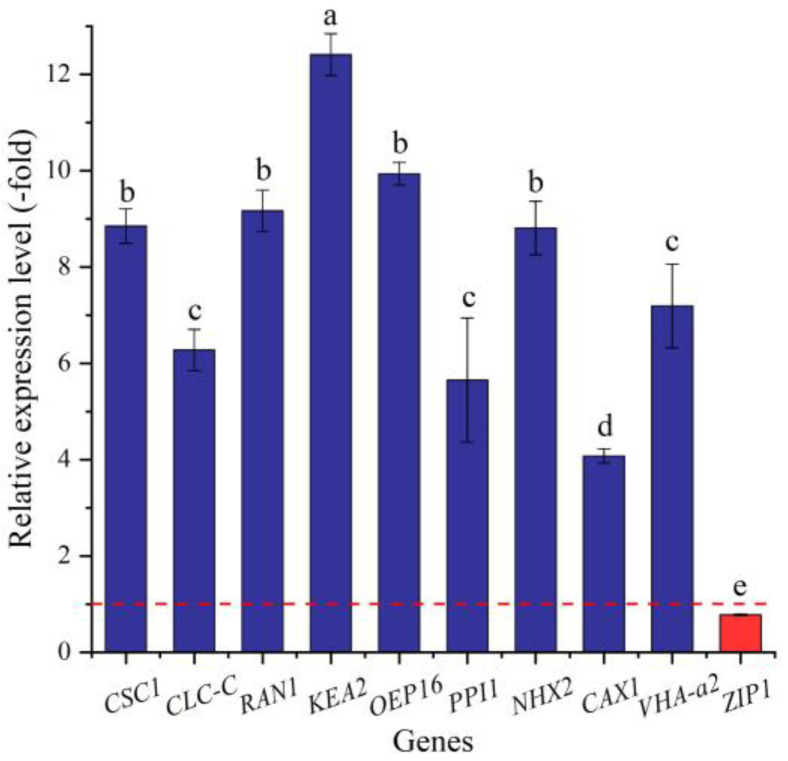
The RELs of genes directly associated with ion transport in *A. venetum* seeds at 300 mmol/L NaCl vs. CK (mean ± SD, *n* = 3). Different letters represent a significant difference (*p* < 0.05) among different genes. The red dotted line differentiates UR (>1) and DR (<1).

**Figure 13 ijms-24-03623-f013:**
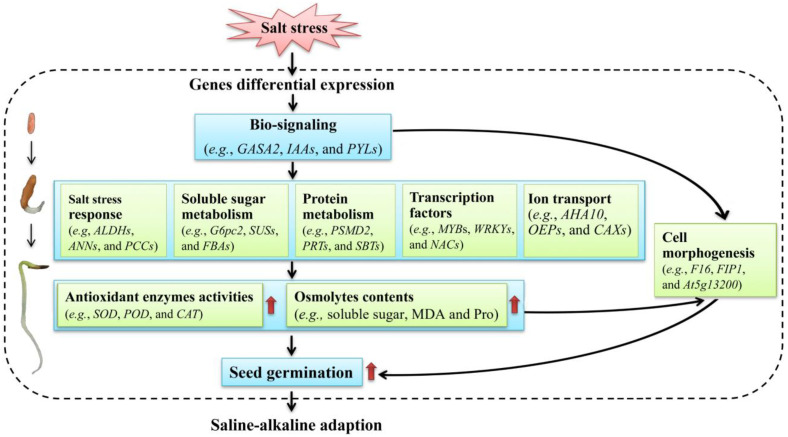
A proposed model of the low-salt-stress-enhanced seed germination of *A. venetum*.

**Table 1 ijms-24-03623-t001:** Summary of sequencing data of seed germination under the CK and 300 mmol/L treatments (mean ± SD, *n* = 3).

	CK	300 mmol/L NaCl
**Filtered data**		
Data of reads number (million)	51.43 ± 12.23	65.36 ± 26.19
Data of reads number×read length (million)	7714 ± 1835	9803 ± 3928
Q20 (%)	97.18 ± 0.28	96.99 ± 0.04
Q30 (%)	92.33 ± 0.56	91.93 ± 0.08
**Mapped data**		
Data of unique mapped reads (million)	42.12 ± 10.02	53.26 ± 21.33
Data of multiple mapped reads (million)	0.81± 0.11	0.80 ± 0.33
Mapping ratio (%)	83.47 ± 3.08	82.71 ± 4.12

**Table 2 ijms-24-03623-t002:** Twenty-nine genes directly associated with salt stress at 300 mmol/L NaCl vs. CK.

Gene Name	SwissProt ID	Protein Name	log_2_FC (NaCl vs. CK)
*ALDH7B4*	Q9SYG7	Aldehyde dehydrogenase family 7 member B4	4.13
*ALDH10A8*	Q9S795	Aminoaldehyde dehydrogenase ALDH10A8	3.91
*ANN2*	Q9XEE2	Annexin D2	4.33
*ANN5*	Q9C9X3	Annexin D5	4.51
*B2*	P37707	B2 protein	4.38
*RD22*	Q08298	BURP domain protein RD22	4.60
*CAMBP25*	O80683	Calmodulin-binding protein 25	4.19
*CTL1*	Q9MA41	Chitinase-like protein 1	3.03
*RD19A*	P43296	Cysteine protease RD19A	3.95
*PCC13*	P22242	Desiccation-related protein PCC13-62	−2.53
*PCC27*	P22241	Desiccation-related protein PCC27-45	3.94
*FLZ13*	Q8GRN0	FCS-Like Zinc finger 13	−2.97
*GRP1*	Q03878	Glycine-rich RNA-binding protein	5.23
*RBG7*	Q03250	Glycine-rich RNA-binding protein 7	3.30
*HMGB2*	O49596	High mobility group B protein 2	3.61
*IQM4*	O64851	IQ domain-containing protein IQM4	8.89
*NRP1*	Q9ZQ80	Nodulin-related protein 1	3.65
*ATP1*	Q9LU63	Probable pterin-4-alpha-carbinolamine dehydratase	4.80
*ARP1*	Q9M1S3	Probable RNA-binding protein ARP1	−1.09
*ERD7*	O48832	Protein EARLY-RESPONSIVE TO DEHYDRATION 7	8.98
*NFD4*	F4I9E1	Protein NUCLEAR FUSION DEFECTIVE 4	4.68
*AVP1*	P31414	Pyrophosphate-energized vacuolar membrane proton pump 1	3.98
*REM4.1*	Q93YN8	Remorin 4.1	1.16
*ALDH5F1*	Q9SAK4	Succinate-semialdehyde dehydrogenase	4.24
*TIL*	Q9FGT8	Temperature-induced lipocalin-1	4.08
*WSD1*	Q93ZR6	Wax ester synthase/diacylglycerol acyltransferase 1	−1.11
*At2g40140*	Q9XEE6	Zinc finger CCCH domain-containing protein 29	1.30
*Os07g0682400*	Q0D3J9	Zinc finger CCCH domain-containing protein 53	3.70
*ZFNL*	Q9SWF9	Zinc finger CCCH domain-containing protein ZFN-like	5.48

**Table 3 ijms-24-03623-t003:** Twenty-three genes directly associated with antioxidant enzymes at the 300 mmol/L NaCl vs. CK.

Gene Name	SwissProt ID	Protein Name	log_2_FC (NaCl vs. CK)
**SOD (3)**			
*SODCP*	P07505	Superoxide dismutase (Cu-Zn), chloroplastic	3.98
*FSD2*	Q9LU64	Superoxide dismutase (Fe) 2, chloroplastic	3.62
*SODA*	P11796	Superoxide dismutase (Mn), mitochondrial	8.98
**POD (17)**			
*At5g06290*	Q9C5R8	2-Cys peroxiredoxin BAS1-like, chloroplastic	3.39
*APX3*	Q42564	L-ascorbate peroxidase 3	3.31
*APX1*	P48534	L-ascorbate peroxidase, cytosolic	1.08
*PER12*	Q96520	Peroxidase 12	5.22
*PER23*	O80912	Peroxidase 23	−3.11
*PER31*	Q9LHA7	Peroxidase 31	5.22
*PER42*	Q9SB81	Peroxidase 42	3.83
*PER52*	Q9FLC0	Peroxidase 52	1.52
*poxN1*	Q9XIV8	Peroxidase N1	−3.93
*PRDX1*	Q06830	Peroxiredoxin-1	9.68
*PRDX2*	A9PCL4	Peroxiredoxin-2	3.52
*PRXIIE*	Q949U7	Peroxiredoxin-2E, chloroplastic	6.02
*PEX5*	Q9FMA3	Peroxisome biogenesis protein 5	3.81
*PEX11C*	Q9LQ73	Peroxisomal membrane protein 11C	4.38
*PEX13*	Q9SRR0	Peroxisomal membrane protein 13	8.99
*PEX14*	Q9FXT6	Peroxisomal membrane protein PEX14	4.29
*GPX1*	P52032	Phospholipid hydroperoxide glutathione peroxidase 1	3.70
**CAT (3)**			
*CATA*	Q9AXH0	Catalase	3.80
*CAT1*	P17598	Catalase isozyme 1	3.78
*PNC1*	P22195	Cationic peroxidase 1	3.52

**Table 4 ijms-24-03623-t004:** Seventy-nine DEGs directly associated with soluble sugar and protein metabolism at 300 mmol/L NaCl vs. CK.

Gene Name	SwissProt ID	Protein Name	log_2_FC (NaCl vs. CK)
**Glucose (11)**			
*G6pc2*	Q9Z186	Glucose-6-phosphatase 2	9.28
*G6PDC*	Q43839	Glucose-6-phosphate 1-dehydrogenase, chloroplastic	3.29
*GAPB*	P12859	Glyceraldehyde-3-phosphate dehydrogenase B	3.73
*GAPA1*	P25856	Glyceraldehyde-3-phosphate dehydrogenase GAPA1	3.85
*GAPC2*	Q9FX54	Glyceraldehyde-3-phosphate dehydrogenase GAPC2	3.65
*PGMP*	Q9M4G5	Phosphoglucomutase, chloroplastic	3.47
*PGM1*	Q9ZSQ4	Phosphoglucomutase, cytoplasmic	3.20
*Gcg*	P55095	Pro-glucagon	12.87
*PSL5*	Q9FN05	Probable glucan 1,3-alpha-glucosidase	3.31
*UGD3*	Q9AUV6	UDP-glucose 6-dehydrogenase 3	3.50
*UGPA*	P19595	UTP--glucose-1-phosphate uridylyltransferase	4.88
**Sucrose (3)**			
*INVE*	Q9FK88	Alkaline/neutral invertase E, chloroplastic	8.90
*SUS2*	O24301	Sucrose synthase 2	3.82
*SUS3*	Q9M111	Sucrose synthase 3	−3.29
**Fructose (9)**			
*FKFBP*	Q9MB58	6-phosphofructo-2-kinase/fructose-2,6-bisphosphatase	3.50
*RHVI2*	H2DF88	Acid beta-fructofuranosidase 2, vacuolar	−1.32
*FRK2*	Q42896	Fructokinase-2	5.37
*FBP*	P46275	Fructose-1,6-bisphosphatase, chloroplastic	3.26
*F16P2*	P46276	Fructose-1,6-bisphosphatase, cytosolic	4.07
*FBA2*	Q944G9	Fructose-bisphosphate aldolase 2, chloroplastic	3.49
*FBA3*	Q9ZU52	Fructose-bisphosphate aldolase 3, chloroplastic	1.93
*FBA6*	Q9SJQ9	Fructose-bisphosphate aldolase 6, cytosolic	1.31
*FBA1*	P46256	Fructose-bisphosphate aldolase, cytoplasmic isozyme 1	4.00
**Galactose (2)**			
*GOLS1*	O22893	Galactinol synthase 1	8.16
*GOLS2*	C7G304	Galactinol synthase 2	8.35
**Trehalose (2)**			
*TPS6*	Q94AH8	Alpha,alpha-trehalose-phosphate synthase	4.20
*TPS7*	Q9LMI0	Probable alpha,alpha-trehalose-phosphate synthase	3.11
**Fucose (2)**			
*OFUT19*	Q9SH89	O-fucosyltransferase 19	1.08
*OFUT39*	Q0WUZ5	O-fucosyltransferase 39	3.78
**Starch (12)**			
*SBEI*	Q41058	1,4-alpha-glucan-branching enzyme 1	3.43
*DPE2*	Q8RXD9	4-alpha-glucanotransferase DPE2	4.63
*AMY3*	Q94A41	Alpha-amylase 3, chloroplastic	3.49
*R1*	Q8LPT9	Alpha-glucan water dikinase, chloroplastic	3.37
*BAM1*	Q9LIR6	Beta-amylase 1, chloroplastic	4.43
*BAM3*	O23553	Beta-amylase 3, chloroplastic	4.61
*ADG2*	P55229	Glucose-1-phosphate adenylyltransferase large subunit 1	3.43
*AGPS1*	Q9M462	Glucose-1-phosphate adenylyltransferase small subunit	2.91
*ISA1*	D0TZF0	Isoamylase 1, chloroplastic	5.35
*DSP4*	G4LTX4	Phosphoglucan phosphatase DSP4, amyloplastic	3.88
*GWD3*	Q6ZY51	Phosphoglucan, water dikinase, chloroplastic	4.48
*SS4*	Q0WVX5	Probable starch synthase 4, chloroplastic/amyloplastic	5.33
**Protein (38)**			
*PSMD2*	Q5R9I6	26S proteasome non-ATPase regulatory subunit 2	4.39
*RPN1A*	Q9SIV2	26S proteasome non-ATPase regulatory subunit 2 homolog A	3.51
*RPN9B*	Q8GYA6	26S proteasome non-ATPase regulatory subunit 13 homolog B	8.52
*RPN10*	P55034	26S proteasome non-ATPase regulatory subunit 4 homolog	4.94
*RPT5A*	Q9SEI2	26S proteasome regulatory subunit 6A homolog A	3.39
*RPT1A*	P0DKJ9	26S proteasome regulatory subunit 7A	5.17
*RPT6A*	Q9C5U3	26S proteasome regulatory subunit 8 homolog A	4.29
*RPT4A*	Q9SEI3	26S proteasome regulatory subunit 10B homolog A	8.74
*PCS1*	Q9LZL3	Aspartic proteinase PCS1	3.89
*APF2*	Q9LNJ3	Aspartyl protease family protein 2	8.63
*At5g10770*	Q8S9J6	Aspartyl protease family protein At5g10770	4.31
*Cys*	Q86GF7	Crustapain	−10.16
*SMAC_06893*	D1ZSU8	Extracellular metalloprotease SMAC_06893	−6.78
*GGP5*	O82225	Gamma-glutamyl peptidase 5	4.19
*LAP2*	Q944P7	Leucine aminopeptidase 2, chloroplastic	3.76
*Pcsk2*	P21661	Neuroendocrine convertase 2	5.62
*maoI*	Q07121	Primary amine oxidase	3.53
*RD19C*	Q9SUL1	Probable cysteine protease RD19C	5.48
*RD21B*	Q9FMH8	Probable cysteine protease RD21B	2.33
*MPPbeta*	Q42290	Probable mitochondrial-processing peptidase subunit beta	6.03
*Prep*	Q9QUR6	Prolyl endopeptidase	−2.29
*DEGP1*	O22609	Protease Do-like 1, chloroplastic	5.36
*DEGP2*	O82261	Protease Do-like 2, chloroplastic	3.67
*PBG1*	Q7DLR9	Proteasome subunit beta type-4	8.91
*PBE2*	Q9LIP2	Proteasome subunit beta type-5-B	4.84
*PBA1*	Q8LD27	Proteasome subunit beta type-6	4.68
*MPA1*	Q8H0S9	Puromycin-sensitive aminopeptidase	3.52
*Rbp3*	P49194	Retinol-binding protein 3	−7.96
*RBL6*	Q8VZ48	RHOMBOID-like protein 6, mitochondrial	5.28
*SCPL49*	P32826	Serine carboxypeptidase-like 49	4.84
*SBT1.2*	O64495	Subtilisin-like protease SBT1.2	−1.73
*SBT1.4*	Q9LVJ1	Subtilisin-like protease SBT1.4	8.46
*SBT1.6*	O49607	Subtilisin-like protease SBT1.6	3.21
*SBT1.7*	O65351	Subtilisin-like protease SBT1.7	0.77
*SBT2.5*	O64481	Subtilisin-like protease SBT2.5	4.43
*SPDS1*	Q96556	Spermidine synthase 1	5.04
*ALEU*	Q8H166	Thiol protease aleurain	3.66
*TPP2*	F4JVN6	Tripeptidyl-peptidase 2	4.06

**Table 5 ijms-24-03623-t005:** Six DEGs directly associated with cell morphogenesis for seed germination at 300 mmol/L NaCl vs. CK.

Gene Name	SwissProt ID	Protein Name	log_2_FC (NaCl vs. CK)
*F16*	Q8W4Z5	CASP-like protein F16	9.63
*FIP1*	Q9SE96	GEM-like protein 1	5.00
*At5g13200*	Q9LYV6	GEM-like protein 5	7.09
*At5g23350*	Q9FMW6	GEM-like protein 6	−1.28
*KAI2*	Q9SZU7	Probable esterase KAI2	9.61
*ROH1*	Q9CAK4	Protein ROH1	3.61

**Table 6 ijms-24-03623-t006:** Sixteen TFs directly associated with stress response and seed germination at 300 mmol/L NaCl vs. CK.

Gene Name	SwissProt ID	Protein Name	log_2_FC (NaCl vs. CK)
**MYB (2)**			
*MYB73*	O23160	Transcription factor MYB73	5.75
*MYB1R1*	Q2V9B0	Transcription factor MYB1R1	3.32
**WRKY (7)**			
*WRKY4*	Q9XI90	Probable WRKY transcription factor 4	4.19
*WRKY23*	O22900	WRKY transcription factor 23	1.15
*WRKY24*	Q6IEQ7	WRKY transcription factor WRKY24	5.42
*WRKY33*	Q8S8P5	Probable WRKY transcription factor 33	4.71
*WRKY40*	Q9SAH7	Probable WRKY transcription factor 40	−1.20
*WRKY49*	Q9FHR7	Probable WRKY transcription factor 49	1.48
*WRKY71*	Q93WV4	WRKY transcription factor 71	2.01
**NAC (4)**			
*NAC083*	Q9FY93	NAC domain-containing protein 83	3.76
*NAC091*	Q9LKG8	NAC domain-containing protein 91	5.15
*NAC100*	Q9FLJ2	NAC domain-containing protein 100	1.34
*JA2L*	A0A3Q7HH64	NAC domain-containing protein JA2L	1.64
**bZIP (1)**			
*BZIP44*	C0Z2L5	bZIP transcription factor 44	5.39
**TCP (1)**			
*TCP14*	Q93Z00	Transcription factor TCP14	3.78
**UNE (1)**			
*UNE10*	Q8GZ38	Transcription factor UNE10	4.16

**Table 7 ijms-24-03623-t007:** Thirty-six DEGs directly associated with hormone response at 300 mmol/L NaCl vs. CK.

Gene Name	SwissProt ID	Protein Name	log_2_FC (NaCl vs. CK)
**GA (1)**			
*GASA2*	P46688	Gibberellin-regulated protein 2	4.35
**IAA (14)**			
*ABP19A*	Q9ZRA4	Auxin-binding protein ABP19a	3.50
*AUX22D*	O24542	Auxin-induced protein 22D	3.32
*AUX12KD*	Q05349	Auxin-repressed 12.5 kDa protein	3.57
*ARF2A*	Q2LAJ3	Auxin response factor 2A	−2.28
*ARF2B*	K4DF01	Auxin response factor 2B	3.98
*ARF6*	Q9ZTX8	Auxin response factor 6	4.48
*IAA4*	P33077	Auxin-responsive protein IAA4	3.62
*IAA8*	Q38826	Auxin-responsive protein IAA8	4.07
*IAA9*	Q38827	Auxin-responsive protein IAA9	4.24
*IAA14*	Q38832	Auxin-responsive protein IAA14	3.85
*SAUR71*	Q9SGU2	Auxin-responsive protein SAUR71	−1.28
*LAX2*	Q9FEL7	Auxin transporter-like protein 2	2.89
*TET4*	Q9LSS4	Tetraspanin-4	4.97
*TET8*	Q8S8Q6	Tetraspanin-8	3.93
**CTK (3)**			
*CISZOG2*	Q8RXA5	Cis-zeatin O-glucosyltransferase 2	−1.09
*AHK3*	Q9C5U1	Histidine kinase 3	4.69
*ARR4*	O82798	Two-component response regulator ARR4	1.09
**JA (2)**			
*JOX2*	Q9FFF6	Jasmonate-induced oxygenase 2	2.21
*TIFY6B*	Q9LVI4	Protein TIFY 6B	3.61
**ABA (6)**			
*PYL3*	Q6EN42	Abscisic acid receptor PYL3	3.73
*PYL4*	O80920	Abscisic acid receptor PYL4	−1.14
*GRDP1*	Q9ZQ47	Glycine-rich domain-containing protein 1	3.49
*PP2CA*	P49598	Protein phosphatase 2C 37	5.90
*PP2C51*	Q65XK7	Protein phosphatase 2C 51	1.43
*SAL1*	Q42546	SAL1 phosphatase	3.31
**ETH (10)**			
*RAV1*	Q9ZWM9	AP2/ERF and B3 domain-containing transcription factor	3.47
*ETR2*	Q0WPQ2	Ethylene receptor 2	−2.03
*ERF4*	Q9LW49	Ethylene-responsive transcription factor 4	9.00
*ERF5*	Q40478	Ethylene-responsive transcription factor 5	0.83
*ERF013*	Q9CAP4	Ethylene-responsive transcription factor ERF013	−2.74
*ERF016*	Q9C591	Ethylene-responsive transcription factor ERF016	2.11
*ERF113*	Q9LYU3	Ethylene-responsive transcription factor ERF113	1.95
*RAP2-2*	Q9LUM4	Ethylene-responsive transcription factor RAP2-2	4.50
*RAP2-4*	Q8H1E4	Ethylene-responsive transcription factor RAP2-4	3.65
*EIN3*	O24606	Protein ETHYLENE INSENSITIVE 3	4.05

**Table 8 ijms-24-03623-t008:** Thirty-four DEGs directly associated with ion transport at 300 mmol/L NaCl vs. CK.

Gene Name	SwissProt ID	Protein Name	log_2_FC (NaCl vs. CK)
*AHA10*	Q43128	ATPase 10, plasma membrane-type	3.98
*CSC1*	Q5XEZ5	Calcium permeable stress-gated cation channel 1	4.65
*CLC-B*	P92942	Chloride channel protein CLC-b	4.68
*CLC-C*	Q96282	Chloride channel protein CLC-c	3.91
*ATX1*	Q94BT9	Copper transport protein ATX1	6.95
*PAA2*	B9DFX7	Copper-transporting ATPase PAA2, chloroplastic	8.69
*RAN1*	Q9S7J8	Copper-transporting ATPase RAN1	4.28
*ERD4*	A9LIW2	CSC1-like protein ERD4	3.81
*KEA2*	O65272	K(+) efflux antiporter 2, chloroplastic	5.20
*MOT1*	Q9SL95	Molybdate transporter 1	−2.18
*NEW1*	Q08972	[NU+] prion formation protein 1	−9.03
*OEP16*	Q41050	Outer envelope pore protein 16, chloroplastic	3.58
*OEP162*	Q0WMZ5	Outer envelope pore protein 16-2, chloroplastic	1.59
*PHO1-H1*	Q93ZF5	Phosphate transporter PHO1 homolog 1	−1.21
*PPI1*	O23144	Proton pump-interactor 1	3.54
*Atp2a3*	Q64518	Sarcoplasmic/endoplasmic reticulum calcium ATPase 3	9.37
*NHX2*	Q56XP4	Sodium/hydrogen exchanger 2	3.07
*ATP1A3*	P13637	Sodium/potassium-transporting ATPase subunit alpha-3	−9.32
*CAX1*	Q39253	Vacuolar cation/proton exchanger 1	2.30
*CAX3*	Q93Z81	Vacuolar cation/proton exchanger 3	3.94
*VPS2.1*	Q9SKI2	Vacuolar protein sorting-associated protein 2 homolog 1	4.27
*VPE*	P49043	Vacuolar-processing enzyme	1.64
*VSR1*	P93026	Vacuolar-sorting receptor 1	4.04
*VATL*	Q96473	V-type proton ATPase 16 kDa proteolipid subunit	4.97
*VHA-a2*	Q9SJT7	V-type proton ATPase subunit a2	4.06
*VATB1*	Q43432	V-type proton ATPase subunit B 1	3.65
*VHA-C*	Q9SDS7	V-type proton ATPase subunit C	5.36
*VHA-D*	Q9XGM1	V-type proton ATPase subunit D	4.70
*VHA-d2*	Q9LHA4	V-type proton ATPase subunit d2	4.43
*VATE*	Q9SWE7	V-type proton ATPase subunit E	4.41
*VHA-H*	Q9LX65	V-type proton ATPase subunit H	4.04
*ZIP1*	O81123	Zinc transporter 1	−1.58
*ZIP4*	O04089	Zinc transporter 4, chloroplastic	−1.06
*ZIP5*	Q6L8G0	Zinc transporter 5	−2.14

## Data Availability

The datasets are publicly available at NCBI (https://www.ncbi.nlm.nih.gov/bioproject/PRJNA847749, accessed on 2 August 2022), and Sequence Read Archive (SRA) accession: CK (SRR20341324, SRR20341330, and SRR20341336), and 300 mmol/L NaCl (SRR20341332, SRR20341333, and SRR20341334).

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
