# Peer review of "Physiological and Transcriptional Responses of *Apocynum venetum* to Salt Stress at the Seed Germination Stage"

_ijms, 2023, doi:10.3390/ijms24043623_

Round 1

Reviewer 1 Report

The manuscript "Physiological and Transcriptional Responses of Apocynum venetum to Salt Stress at Seed Germination Stage" presents a large set of quality experimental data.

However, it seems unclear that the expression changes shown for high 300 mM NaCl  explain the stimulation of seed germination of proposed model for low salt concentration.

It's not totally clear what 40 independent biological replicates means - 40 Petri dishes or what?

What is "three biological replicates (a total of 120 samples = 3 seeds × 40 independent biological replicates") means?

It  could be interesting to highlight the genes with altered expression under salt stress which are responsible for specific responses to saline stress, as the majority of mentioned genes provide nonspecific reactions. to any kind of stress.

Author Response

Thanks very much for your comments that are helpful to improve our paper much better. We have tried to address and correct each comment. Attachments below with our responses are shown in bold. Revised parts are highlighted with “track change” in the manuscript. If any question remains, please let us know.

Reviewer 1 comments:

1. The manuscript "Physiological and Transcriptional Responses of Apocynum venetum to Salt Stress at Seed Germination Stage" presents a large set of quality experimental data. However, it seems unclear that the expression changes shown for high 300 mM NaCl explain the stimulation of seed germination of proposed model for low salt concentration.

Thanks very much for your positive comments. Indeed, the transcriptomic analysis should be conducted on the 50 mM NaCl (improve the seed germination) vs. CK (0 mM) to explain the stimulation of seed germination of proposed model for low salt concentration. In this study, in order to reveal the adaptive mechanism of A. venetum to salt stress, especially under higher salt stress, the gene expression changes (i.e., transcriptomic analysis) were conducted on 300 mM NaCl vs. CK, which maybe favor of digging out more genes involved in salt tolerance.

According to your comments, a description: “In order to find our far more genes involved in salt tolerance, transcriptomic analysis was conducted on higher salt concentration (300 mM) instead of lower salt concentration (50) vs. CK.” in the manuscript. (Page 4, lines 158-160) 

2. It's not totally clear what 40 independent biological replicates means - 40 Petri dishes or what?

The 40 independent biological replicates means 40 Petri dishes, the description: “Each treatment has 40 independent biological replicates, which were germinated in 40 Petri-dishs (30 seeds per dish).” has been added in the text. (Page 19, lines 685-686)

3. What is "three biological replicates (a total of 120 samples = 3 seeds × 40 independent biological replicates") means?

The “three biological replicates (a total of 120 samples = 3 seeds × 40 independent biological replicates)” means “Each determination contains three biological replicates, each biological replicate contains 40 Petri dishes with 3 seeds.” (Page 19, lines 685-686)

4. It could be interesting to highlight the genes with altered expression under salt stress which are responsible for specific responses to saline stress, as the majority of mentioned genes provide nonspecific reactions to any kind of stress.

According to your comments, the specific functions of genes associated with salt stress have been highlight provided: “ALDH7B4 is differentially activated by high salinity, dehydration, and ABA in a tissue-specific manner, and involves diverged signal transduction pathways [48]; Annexin (ANNs) genes in Arabidopsis is differentially regulated by exposure to salt, drought, and high- and low-temperature conditions [49]; RD22 is induced by salt stress as well as by water deficit, and expressed during the early and middle stages of seed development [50]. CAMBP25 is encoded by a single-copy gene, whose expression is induced in Arabidopsis seedlings exposed to high salinity, dehydration, and low temperature [51]. PCC13 is abundantly expressed in dried leaves and abscisic acid-treated dried callus, and involved in response to salt stress [52].” (Page 16, lines 377-383)

Reviewer 2 Report

Discussion: It should be more precise and informative. It seems very clumsy. Rewrite the section with latest references.

Author Response

Thanks very much for your comments that are helpful to improve our paper much better. We have tried to address and correct each comment. Attachments below with our responses are shown in bold. Revised parts are highlighted with “track change” in the manuscript. If any question remains, please let us know.

Reviewer 2 comments:

The author’s investigated the physiological and transcriptional changes during seed germination at different NaCl treatments (0-300 mmol/L) in A. venetum in response to salt stress. The transcriptional studies shows a series of genes either up-regulated or down regulated in repose to salinity levels. The manuscript sounds scientific and will provide useful references to improve the seed germination and reveal the adaptive mechanism of A. venetum to saline–alkaline soils. However some points are suggested to improve the overall quality of the manuscript before final publication.

1. Moderate English editing is required and some typographical errors must be corrected.

According to your comments, the English languade and typographical errors have been carefully checked and corrected throughout the text.

Suggestions for authors:

2. In Figure 4c, the cluster heat map of DEGs is confusing. Justify it.

The information about the cluster heat map of DEGs has been provided: “The cluster heat map of the 1967 DEGs was shown in Figure 4C, the distinct difference in expression level of genes between NaCl and CK treatments indicates that the data can be used for further analysis of gene biological functions.” (Page 5, lines 195-197)

3. In qPCR studies, what was the reference gene? How you have calculated the fold change in gene expression.

In this study, the Actin gene was used as a reference based on our previously published article: “Wu, Z.B.; Chang, P.X.; Zhao, J.; Wang, W.S.; Cui, X.W.; Li, M.F. Physiological and transcriptional responses of seed germination to moderate drought in Apocynum venetum. Front. Ecol. Evol. 2022, 10, 975771.” (Page 21, lines 775; Page 22, lines 877-878)

Additionally, the specific method of calculated the fold change in gene expression has been added to the text: “The RELs of genes were calculated using a 2△△Ct method (Ct, Cycle threshold value of target gene) according to the following formula.” (Page 21, lines 776-777)

Ct Test gene=Ct Test geneCt Reference gene

Ct Control gene =Ct Control geneCt Reference gene

△△Ct=(Ct Test geneCt Control gene)

Relative gene expression fold (Test gene/Control gene)=2-△△Ct

4. Results: The physiological and biochemical traits should be elaborated.

According to your comments, the results of “section 2.1. Seed Germination Rate at Different NaCl Treatments”, “section 2.2. Antioxidant Enzymes Activities at Different NaCl Treatments”, and “section 2.3. Osmolytes Contents at Different NaCl Treatments” have been revised. (Page 2, lines 94-97; Page 3, lines 129-134 and 140-146)

5. Discussion: It should be more precise and informative. It seems very clumsy. Rewrite the section with latest references.

Thanks for your suggestion, the Discussion section has been rewrited, such as:

(1) Saline and alkaline ions can reduce the soil water potential, which makes it difficult for plants to absorb water, and then causes osmotic stress [35]. Previous studies have demonstrated that genes related to soluble sugar and protein metabolism paly critical roles in response to salt stress in other plants. For example, the trehalose-6-phosphate synthase (TPS) gene plays the critical role in Citrullus lanatus response to salt stress [64]; OsGolS1 gene significantly up-regulated in Oryza sativa ssp. japonica under salt stress [65]; The overexpression of MsTRX in tobacco induced the upregulation of beta-amylase 1 (BAM1) under salt stress [66]. (Page 16, lines 407-414)

(2) TFs are emerged as key regulators in various signaling networks and play significant roles by improving the growth and development of plants under stress conditions [90]. Previous studies have demonstrated that genes related to TFs paly critical roles in response to salt stress in other plants. For example, WRKY TFs of cotton play a significant roles in the regulation of abiotic stresses (i.e., salt, drought, and extreme temperatures) [91]; MYB TFs involved in the responses to different abiotic stresses, such as salt, cold, and drought, and FvMYB82 gene probably plays an important role in the response to salt and cold stresses in Arabidopsis thaliana by regulating downstream related genes [92]; NAC TFs play vital roles in plant development and responses to various abiotic stresses, and ThNAC4 gene of Tamarix hispida involved in salt and osmotic stress tolerance [93]. (Page 17, lines 498-508)

(3) As is known, endogenous hormones play critical roles in plant growth and development. Previous studies have demonstrated that genes related to hormone-signinaling paly critical roles in response to salt stress in other plants. For example, SmGASA4 was found to be positively regulated by Gibberellin (GA) and significantly enhanced plant resistance to salt, drought, and paclobutrazol (PBZ) stress in Salvia miltiorrhiza [101]; Aux/IAA proteins in auxin and ARF transcription factors directly regulate auxin-responsive gene expression, and OsIAA24 and OsIAA20 are up-regulated in rice under high salt stress [102]; Genes related to Cytokinin (CK) and ethylene (ET) were involved in alleviating the root damage of Tamarix ramosissima under NaCl stress [103]. (Page 18, lines 574-582)

(4) Saline and alkaline stress can induce ion toxicity to plants when too many toxic ions enter plant cells, which can harm the plant cytoplasm and organelles; among them, Na+ is the main toxicity ion due to the similarity in size of the hydrated ionic radii of Na+ and K+, which makes them difficult to be discriminated [115]. (Page 18, lines 599-602)

(5) Previous studies have demonstrated that genes related to ion transport paly critical roles in response to salt stress in other plants. For example, CLC-c is involved in response to salt stress tolerance and seed germination in Gossypium hirsutum [117]; K+ efflux transporters (KEAs) were expressed under abiotic stress (salt, heat, and drought) in Cajanus cajan [118]; Zipper (Zip) gene family participates in plant growth and development and ability to cope with outside environment stresses, which may potentially regulate seed germination and stress resistance in Miscanthus sinensis [119]”. (Page 18, lince 604-611)

  1. Liu, D.; Ma, Y.; Rui, M.M.; Lv, X.C.; Chen, R.J.; Chen, X.Y.; Wang, Y.Z. Is high pH the key factor of alkali stress on plant growth and physiology? A case study with Wheat (Triticum aestivum L.) seedlings. Agronomy 2022, 12, 1820.
  2. Yuan, G.; Liu, J.; An, G.; Li, W.; Si, W.; Sun, D.; Zhu, Y. Genome-wide identification and characterization of the trehalose-6-phosphate synthetase (TPS) gene family in Watermelon (Citrullus lanatus) and their transcriptional responses to salt stress. Int. J. Mol. Sci. 2022, 23, 276.
  3. Kong, W.; Gong, Z.; Zhong, H.; Zhang, Y.; Zhao, G.; Gautam, M.; Deng, X.; Liu, C.; Zhang, C.; Li, Y. Expansion and evolutionary patterns of glycosyltransferase family 8 in gramineae crop genomes and their expression under salt and cold stresses in Oryza sativa ssp. japonica. Biomolecules 2019, 9, 188.
  4. Duan, X.; Wang, Z.; Zhang, Y.; Li, H.; Yang, M.; Yin, H.; Cui, J.; Chai, H.; Gao, Y.; Hu, G.; Zhang, P. Overexpression of a thioredoxin-protein-encoding gene, MsTRX, from Medicago sativa enhances salt tolerance to transgenic tobacco. Agronomy 2022, 12, 1467.
  5. Cheng, C.; An, L.; Li, F.; Ahmad, W.; Aslam, M.; Ul Haq, M.Z.; Yan, Y.; Ahmad, R.M. Wide-range portrayal of AP2/ERF transcription factor family in Maize (Zea mays L.) development and stress responses. Genes 2023, 14, 194.
  6. Guo, X.; Ullah, A.; Siuta, D.; Kukfisz, B.; Iqbal, S. Role of WRKY transcription factors in regulation of abiotic stress responses in Cotton. Life 2022, 12, 1410.
  7. Li, W.; Zhong, J.; Zhang, L.; Wang, Y.; Song, P.; Liu, W.; Li, X.; Han, D. Overexpression of a Fragaria vesca MYB transcription factor gene (FvMYB82) increases salt and cold tolerance in Arabidopsis thaliana. Int. J. Mol. Sci. 2022, 23, 10538.
  8. Mijiti, M.; Wang, Y.; Wang, L.; Habuding, X. Tamarix hispida NAC transcription factor ThNAC4 confers salt and drought stress tolerance to transgenic Tamarix and Arabidopsis. Plants 2022, 11, 2647.
  9. Wang, H.; Wei, T.; Wang, X.; Zhang, L.; Yang, M.; Chen, L.; Song, W.; Wang, C.; Chen, C. Transcriptome analyses from mutant Salvia miltiorrhiza reveals important roles for SmGASA4 during plant development. Int. J. Mol. Sci. 2018, 19, 2088.
  10. Jain, M.; Khurana, J.P. Transcript profiling reveals diverse roles of auxin-responsive genes during reproductive development and abiotic stress in rice. FEBS J. 2009, 276, 3148–3162.
  11. Chen, Y.; Zhang, S.; Du, S.; Wang, G.; Zhang, J.; Jiang, J. Effects of exogenous (K+) potassium application on plant hormones in the roots of Tamarix ramosissima under NaCl stress. Genes 2022, 13, 1803.
  12. Blumwald, E. Sodium transport and salt tolerance in plants. Curr. Opin. Cell Biol. 2000, 12, 431–434.
  13. Ren, W.; Wang, Q.; Chen, L.; Ren, Y. Transcriptome and metabolome analyses of salt stress response in Cotton (Gossypium hirsutum) seed pretreated with NaCl. Agronomy 2022, 12, 1849.
  14. Siddique, M.H.; Babar, N.I.; Zameer, R.; Muzammil, S.; Nahid, N.; Ijaz, U.; Masroor, A.; Nadeem, M.; Rashid, M.A.R.; Hashem, A.; et al. Genome-wide identification, genomic organization, and characterization of potassium transport-related genes in Cajanus cajan and their role in abiotic stress. Plants 2021, 10, 2238.
  15. Chen, J.; Ran, Q.; Yang, Z.; Zhou, Y.; Yuan, Z.; Lai, H.; Wang, J.; Nie, G.; Zhu, Y. Genome-wide identification and expression profile of the HD-Zip transcription factor family associated with seed germination and abiotic stress response in Miscanthus sinensis. Genes 2022, 13, 2256.

6. What standard protocols the authors have followed to prepare different saline solutions.

According to your comments, the information of standard protocols of prepare different saline solutions has been added to materials: “these doses were selected based on the published literature [11, 16, 17]”. (Page 19, line 684)

7. Rectify spacing error throughout the manuscript.

The spacing error has been rectified throughout the manuscript, such as page 2, lines 58, 63 and 68; page 16, line 404.

8. Conclusion: It should highlight only the major findings of the present study. Rewrite the section.

According to your comments, the conclusion has been rewrited to “This research reveals that salt stress affects the seed germination of A. venetum. The seed germination of A. venetum is promoted at low NaCl concentration such as 50 mmol/L , and significant changes of antioxidant enzymes activities, osmolytes contents, and genes expression levels in A. venetum play critical roles in regulating seed germination under different salt stresses. These findings indicate that A. venetum plants can adapt to salt stress in the saline–alkaline soils by integrating physiological and transcriptional responses. The specific roles of key genes in conferring the ability to salt resistance will require further investigation.” (Page 21, lines 789-796)

Reviewer 3 Report

Dear Authors,

I have an opportunity to review manuscript entitled: “Physiological and Transcriptional Responses of Apocynum venetum to Salt Stress at Seed Germination Stage”

Authors concentrated on physiological and transcriptional changes during seed germination at different NaCl treatments (0-300 mmol/L);

I think that the most strengths of the manuscript is identification1967 DEGs were generated during seed germination at the 300 mmol/L NaCl vs. CK, with 1487 characterized genes; Moreover, the relative expression levels (RELs) of selected genes directly involved in salt stress and seed germination were observed to be consistent with the changes of antioxidant enzymes activities and osmolytes contents.

In my opinion the introduction gives sufficient background for the reader;

Figure 5 and 4 should be enlarge a bit to make the data more visible;

The research is designed appropriate, but I suggest to the all relative expression levels in charts add the statistical significance markings- Figure 6,7,8,10,11,12;

I warmly suggest clearly presented aim of the study, because in current form we have right away what was investigated, what kind of analyses were done;

The material and  methods are clearly described;

Discussion part is an informative part of the manuscript;

The figure 13 - A proposed model of low salt stress enhanced seed germination of A. venetum- was the very good and informative idea, but I suggest to enlarge the figure, because in current form some information are difficult to read;

 English language need some improvements;

Furthermore, the conclusion are very short, therefore I suggest to add some future prospect coming from obtained results to make the results more visible to wider audience;

English language needs some improvements and corrections;

 Sincerely

Author Response

Thanks very much for your comments that are helpful to improve our paper much better. We have tried to address and correct each comment. Attachments below with our responses are shown in bold. Revised parts are highlighted with “track change” in the manuscript. If any question remains, please let us know.

Reviewer 3 comments:

I have an opportunity to review manuscript entitled: “Physiological and Transcriptional Responses of Apocynum venetum to Salt Stress at Seed Germination Stage”. Authors concentrated on physiological and transcriptional changes during seed germination at different NaCl treatments (0-300 mmol/L); I think that the most strengths of the manuscript is identification 1967 DEGs were generated during seed germination at the 300 mmol/L NaCl vs. CK, with 1487 characterized genes; Moreover, the relative expression levels (RELs) of selected genes directly involved in salt stress and seed germination were observed to be consistent with the changes of antioxidant enzymes activities and osmolytes contents.

1. In my opinion the introduction gives sufficient background for the reader.

Thanks very much for your reviewing.

2. Figure 5 and 4 should be enlarge a bit to make the data more visible.

The Figure 5 and 4 should be enlarged more visible, when the Editor office deals with these Figures.   

3. The research is designed appropriate, but I suggest to the all relative expression levels in charts add the statistical significance markings- Figure 6, 7, 8, 10, 11, 12.

According to your comments, the statistical significance has been added to the Figure 6, 7, 8, 9, 10, 11, and 12. (Page 7, lines 229-231; Page 8, lines 245-247; Page 10, lines 265-268; Page 11, lines 282-285; Page 12, lines 299-302; Page 14. lines 318-320; Page 15, lines 332-334)

4. I warmly suggest clearly presented aim of the study, because in current form we have right away what was investigated, what kind of analyses were done.

According to your comments, the aim of the study has been described as “In order to reveal the adaptive mechanism of A. venetum to salt stress, the changes into the rate of seed germination, activity of antioxidant enzymes, content of osmolytes, and expression level of genes in A. venetum at seed germination stage under different NaCl treatments were examined.” (Page 2, lines 88-91)

5. The material and methods are clearly described.

The material and methods have been clearly described “section 4.1. Plant Materials”, “section 4.2 Measurement of Germination Rate”, “section 4.3. Determination of Antioxidant Enzyme Activities”, and “section 4.4. Determination of Osmolytes Content.” (Page 19, lines 677-686 and 688-690; Page 20, lines 702-714 and 716-728)

6. Discussion part is an informative part of the manuscript.

According to your and other reviewers’ comments, the discussion section has been rewrited, such as:

(1) Saline and alkaline ions can reduce the soil water potential, which makes it difficult for plants to absorb water, and then causes osmotic stress [35]. Previous studies have demonstrated that genes related to soluble sugar and protein metabolism paly critical roles in response to salt stress in other plants. For example, the trehalose-6-phosphate synthase (TPS) gene plays the critical role in Citrullus lanatus response to salt stress [64]; OsGolS1 gene significantly up-regulated in Oryza sativa ssp. japonica under salt stress [65]; The overexpression of MsTRX in tobacco induced the upregulation of beta-amylase 1 (BAM1) under salt stress [66]. (Page 16, lines 407-414)

(2) TFs are emerged as key regulators in various signaling networks and play significant roles by improving the growth and development of plants under stress conditions [90]. Previous studies have demonstrated that genes related to TFs paly critical roles in response to salt stress in other plants. For example, WRKY TFs of cotton play a significant roles in the regulation of abiotic stresses (i.e., salt, drought, and extreme temperatures) [91]; MYB TFs involved in the responses to different abiotic stresses, such as salt, cold, and drought, and FvMYB82 gene probably plays an important role in the response to salt and cold stresses in Arabidopsis thaliana by regulating downstream related genes [92]; NAC TFs play vital roles in plant development and responses to various abiotic stresses, and ThNAC4 gene of Tamarix hispida involved in salt and osmotic stress tolerance [93]. (Page 17, lines 498-508)

(3) As is known, endogenous hormones play critical roles in plant growth and development. Previous studies have demonstrated that genes related to hormone-signinaling paly critical roles in response to salt stress in other plants. For example, SmGASA4 was found to be positively regulated by Gibberellin (GA) and significantly enhanced plant resistance to salt, drought, and paclobutrazol (PBZ) stress in Salvia miltiorrhiza [101]; Aux/IAA proteins in auxin and ARF transcription factors directly regulate auxin-responsive gene expression, and OsIAA24 and OsIAA20 are up-regulated in rice under high salt stress [102]; Genes related to Cytokinin (CK) and ethylene (ET) were involved in alleviating the root damage of Tamarix ramosissima under NaCl stress [103]. (Page 18, lines 574-582)

(4) Saline and alkaline stress can induce ion toxicity to plants when too many toxic ions enter plant cells, which can harm the plant cytoplasm and organelles; among them, Na+ is the main toxicity ion due to the similarity in size of the hydrated ionic radii of Na+ and K+, which makes them difficult to be discriminated [115]. (Page 18, lines 599-602)

(5) Previous studies have demonstrated that genes related to ion transport paly critical roles in response to salt stress in other plants. For example, CLC-c is involved in response to salt stress tolerance and seed germination in Gossypium hirsutum [117]; K+ efflux transporters (KEAs) were expressed under abiotic stress (salt, heat, and drought) in Cajanus cajan [118]; Zipper (Zip) gene family participates in plant growth and development and ability to cope with outside environment stresses, which may potentially regulate seed germination and stress resistance in Miscanthus sinensis [119]”. (Page 18, lince 604-611)

  1. Liu, D.; Ma, Y.; Rui, M.M.; Lv, X.C.; Chen, R.J.; Chen, X.Y.; Wang, Y.Z. Is high pH the key factor of alkali stress on plant growth and physiology? A case study with Wheat (Triticum aestivum L.) seedlings. Agronomy 2022, 12, 1820.
  2. Yuan, G.; Liu, J.; An, G.; Li, W.; Si, W.; Sun, D.; Zhu, Y. Genome-wide identification and characterization of the trehalose-6-phosphate synthetase (TPS) gene family in Watermelon (Citrullus lanatus) and their transcriptional responses to salt stress. Int. J. Mol. Sci. 2022, 23, 276.
  3. Kong, W.; Gong, Z.; Zhong, H.; Zhang, Y.; Zhao, G.; Gautam, M.; Deng, X.; Liu, C.; Zhang, C.; Li, Y. Expansion and evolutionary patterns of glycosyltransferase family 8 in gramineae crop genomes and their expression under salt and cold stresses in Oryza sativa ssp. japonica. Biomolecules 2019, 9, 188.
  4. Duan, X.; Wang, Z.; Zhang, Y.; Li, H.; Yang, M.; Yin, H.; Cui, J.; Chai, H.; Gao, Y.; Hu, G.; Zhang, P. Overexpression of a thioredoxin-protein-encoding gene, MsTRX, from Medicago sativa enhances salt tolerance to transgenic tobacco. Agronomy 2022, 12, 1467.
  5. Cheng, C.; An, L.; Li, F.; Ahmad, W.; Aslam, M.; Ul Haq, M.Z.; Yan, Y.; Ahmad, R.M. Wide-range portrayal of AP2/ERF transcription factor family in Maize (Zea mays L.) development and stress responses. Genes 2023, 14, 194.
  6. Guo, X.; Ullah, A.; Siuta, D.; Kukfisz, B.; Iqbal, S. Role of WRKY transcription factors in regulation of abiotic stress responses in Cotton. Life 2022, 12, 1410.
  7. Li, W.; Zhong, J.; Zhang, L.; Wang, Y.; Song, P.; Liu, W.; Li, X.; Han, D. Overexpression of a Fragaria vesca MYB transcription factor gene (FvMYB82) increases salt and cold tolerance in Arabidopsis thaliana. Int. J. Mol. Sci. 2022, 23, 10538.
  8. Mijiti, M.; Wang, Y.; Wang, L.; Habuding, X. Tamarix hispida NAC transcription factor ThNAC4 confers salt and drought stress tolerance to transgenic Tamarix and Arabidopsis. Plants 2022, 11, 2647.
  9. Wang, H.; Wei, T.; Wang, X.; Zhang, L.; Yang, M.; Chen, L.; Song, W.; Wang, C.; Chen, C. Transcriptome analyses from mutant Salvia miltiorrhiza reveals important roles for SmGASA4 during plant development. Int. J. Mol. Sci. 2018, 19, 2088.
  10. Jain, M.; Khurana, J.P. Transcript profiling reveals diverse roles of auxin-responsive genes during reproductive development and abiotic stress in rice. FEBS J. 2009, 276, 3148–3162.
  11. Chen, Y.; Zhang, S.; Du, S.; Wang, G.; Zhang, J.; Jiang, J. Effects of exogenous (K+) potassium application on plant hormones in the roots of Tamarix ramosissima under NaCl stress. Genes 2022, 13, 1803.
  12. Blumwald, E. Sodium transport and salt tolerance in plants. Curr. Opin. Cell Biol. 2000, 12, 431–434.
  13. Ren, W.; Wang, Q.; Chen, L.; Ren, Y. Transcriptome and metabolome analyses of salt stress response in Cotton (Gossypium hirsutum) seed pretreated with NaCl. Agronomy 2022, 12, 1849.
  14. Siddique, M.H.; Babar, N.I.; Zameer, R.; Muzammil, S.; Nahid, N.; Ijaz, U.; Masroor, A.; Nadeem, M.; Rashid, M.A.R.; Hashem, A.; et al. Genome-wide identification, genomic organization, and characterization of potassium transport-related genes in Cajanus cajan and their role in abiotic stress. Plants 2021, 10, 2238.
  15. Chen, J.; Ran, Q.; Yang, Z.; Zhou, Y.; Yuan, Z.; Lai, H.; Wang, J.; Nie, G.; Zhu, Y. Genome-wide identification and expression profile of the HD-Zip transcription factor family associated with seed germination and abiotic stress response in Miscanthus sinensis. Genes 2022, 13, 2256.

7. The figure 13 - A proposed model of low salt stress enhanced seed germination of A. venetum- was the very good and informative idea, but I suggest to enlarge the figure, because in current form some information are difficult to read.

The Figure 13 should be enlarged more visible, when the Editor office deals with it.

8. English language need some improvements.

According to your comments, the English languade and typographical errors have been carefully checked and corrected throughout the text.

9. Furthermore, the conclusion are very short, therefore I suggest to add some future prospect coming from obtained results to make the results more visible to wider audience.

According to your and other reviewer comments, the conclusion has been rewrited to “This research reveals that salt stress affects the seed germination of A. venetum. The seed germination of A. venetum is promoted at low NaCl concentration such as 50 mmol/L , and significant changes of antioxidant enzymes activities, osmolytes contents, and genes expression levels in A. venetum play critical roles in regulating seed germination under different salt stresses. These findings indicate that A. venetum plants can adapt to salt stress in the saline–alkaline soils by integrating physiological and transcriptional responses. The specific roles of key genes in conferring the ability to salt resistance will require further investigation.” (Page 21, lines 789-796)

Reviewer 4 Report

The present study entitled “Physiological and Transcriptional Responses of Apocynum venetum to Salt Stress at Seed Germination Stage” demonstrates the effect of different NaCl treatments (0-300 mmol/L) on physiological and transcriptional changes in Apocynum venetum at seed germination stage. In this study, the activity of antioxidant enzymes and osmolytes along with transcriptomic analysis were performed to examine the seed germination rate in response to different NaCl concentrations in order to find the adaptive immunity of Apocynum venetum under salt stress. The study matter is interesting and of potential interest to the readership of the journal. However, the main concerns about this study can be found below, which need to be addressed and the major revision is suggested.

The study involves several abbreviations, please make sure all the abbreviations have been described at their first mention. Overall language and description of the study needs to be improved.

Page 2, line 49: …temperatures (10/25 and 15/30°C, 12 h/12 h) were more in favor of…

Page 2, line 54: … the activity of antioxidant enzymes…

Page 16, line 306: … in response to drought, salt, and ABA…

Page 16, line 322: … the genes encoding SOD, POD, and CAT…

Page 17, line 357: … seed germination and play a novel role in…

Page 17, lines 391-392: “Under salt stress, high concentrations of… disruption of ion homeostasis.” Please re-write this sentence.

Page 18, line 423: “Briefly, when seeds exposed to salt stress, the gene regulatory will be triggered”. Please re-write this sentence.

Page 19, line 434: “Each treatment has 40 independent biological replicates.” This statement requires more explanation. What are those 40 independent biological replicates and by how those replications were made?

Page 19, line 474: Clean reads were assembled…

Author Response

Thanks very much for your comments that are helpful to improve our paper much better. We have tried to address and correct each comment. Attachments below with our responses are shown in bold. Revised parts are highlighted with “track change” in the manuscript. If any question remains, please let us know.

Reviewer 4 comments:

The present study entitled “Physiological and Transcriptional Responses of Apocynum venetum to Salt Stress at Seed Germination Stage” demonstrates the effect of different NaCl treatments (0-300 mmol/L) on physiological and transcriptional changes in Apocynum venetum at seed germination stage. In this study, the activity of antioxidant enzymes and osmolytes along with transcriptomic analysis were performed to examine the seed germination rate in response to different NaCl concentrations in order to find the adaptive immunity of Apocynum venetum under salt stress. The study matter is interesting and of potential interest to the readership of the journal. However, the main concerns about this study can be found below, which need to be addressed and the major revision is suggested.

1. The study involves several abbreviations, please make sure all the abbreviations have been described at their first mention.

Thanks for your comments, all abbreviations have been described: “differentially expressed genes (DEGs)”; “versus (vs.)”; “Kyoto Encyclopedia of Genes and Genomes (KEGG, 34,198), Eukaryotic Orthologous Groups of proteins (KOG, 22,630), NCBI non-redundant protein (Nr, 36,077)”, as well as Gene Ontology (GO) at their first mention. (Page 1, lines 22 and 23; Page 4, lines 164, 165, 166 and 174).

2. Overall language and description of the study needs to be improved.

According to your comments, the English languade and typographical errors have been carefully checked and corrected throughout the text.

3. Page 2, line 49: …temperatures (10/25 and 15/30°C, 12 h/12 h) were more in favor of…

The sentence of “the fluctuated temperatures (10/25 and 15/30◦C, 12 h/12 h) was more in favor of the seed germination than other temperatures” has been revised to “the fluctuated temperatures (10/25 and 15/30◦C, 12 h/12 h) were more in favor of the seed germination than other temperatures.” (Page 2, line 58)

4. Page 2, line 54: … the activity of antioxidant enzymes…

The sentence of “the activitie of antioxidant enzymes” has been revised to “the activity of antioxidant enzymes.” (Page 2, line 63)

5. Page 16, line 306: … in response to drought, salt, and ABA…

The second drought has been deleted in the text. (Page 16, line 388)

6. Page 16, line 322: … the genes encoding SOD, POD, and CAT…

The sentence of “most of the genes encode SOD, POD, and CAT were UR” has been revised to “most of the genes encoding SOD, POD, and CAT were UR.” (Page 16, line 404)

7. Page 17, line 357: … seed germination and play a novel role in…

The sentence of “GEM-like proteins (i.e., FIP1, At5g13200, and At5g23350) are involved in seed germination and plays a novel role in regulating the reproductive development of plants” has been revised to “GEM-like proteins (i.e., FIP1, At5g13200, and At5g23350) are involved in seed germination and play a novel role in regulating the reproductive development of plants.” (Page 17, line 494)

8. Page 17, lines 391-392: “Under salt stress, high concentrations of… disruption of ion homeostasis.” Please re-write this sentence.

The sentence of “Under salt stress, high concentrations of… disruption of ion homeostasis” has been revised to “Saline and alkaline stress can induce ion toxicity to plants when too many toxic ions enter plant cells, which can harm the plant cytoplasm and organelles; among them, Na+ is the main toxicity ion due to the similarity in size of the hydrated ionic radii of Na+ and K+, which makes them difficult to be discriminated.” (Page 18, lines 599-602)

9. Page 18, line 423: “Briefly, when seeds exposed to salt stress, the gene regulatory will be triggered”. Please re-write this sentence.

The sentence of “Briefly, when seeds were stimulated by salt stress, the gene regulatory will be performed and related genes will be differentially expressed” (Page 19, lines 662-663)

10. Page 19, line 434: “Each treatment has 40 independent biological replicates.” This statement requires more explanation. What are those 40 independent biological replicates and by how those replications were made?

Thanks for your comments, the information of 40 independent has been described to “Each treatment has 40 independent biological replicates, which were made in 40 Petri-dishs (30 seeds per dish).” (Page 19, lines 685-686)

11. Page 19, line 474: Clean reads were assembled…

The sentenced of “Clean reads was assembled…” has been revised to “Clean reads were assembled…”(Page 20, line 744)

Round 2

Reviewer 4 Report

Authors have considerably improved the manuscript entitled “Physiological and Transcriptional Responses of Apocynum venetum to Salt Stress at Seed Germination Stage”. However, the manuscript still involves many concerns, which need to be resolved before publication of this manuscript.

 The description of some results is very complex. It needs a considerable improvement to become easy to understand for readers. Language of the manuscript also needs to be improved, as it involves several grammatical, spelling, and punctuation mistakes.

 Page 2, line 59: “…while the DR of genes involved in cell wall structure”. This statement is not clear, please re-write it.

Page 2, line 74: It should be up regulated.

Page, line 81: “while the adaptive mechanism to salt stress is still limited”. Please do not remove this statement, it should be included.

Page 2, lines 81-85: “In order to reveal the adaptive…treatments were examined.” Please re-write this sentence to “In order to reveal the adaptive mechanism of A. venetum to salt stress in this study, we examined the changes into the rate of seed germination, activity of antioxidant enzymes, content of osmolytes, and expression level of genes in A. venetum at seed germination stage under different NaCl treatments.

Page 2, lines 89-90: It should be “low NaCl concentration (50 mmol/L)…”

Page 3, lines 107-109: …significant changes in osmolyte contents (i.e., soluble sugar, protein, MDA, and Pro) was observed in seeds at different NaCl treatments, with a significant…

Page 3, line 112: mmol/L. While a significant decrease was observed by increasing the NaCl concentrations…

Page 4, line 120:  In order to find far more genes…

Page 5, line 146: …in Figure 4C. The distinct difference in…

Page 6, line 166: It should be “Specific Classification of DEGs and Validation of Expression Levels”

Page 13, line 241: … 8 selected genes were validated…

Page 19, line 312: … low NaCl concentration (50 mmol/L)…

Page 19, line 343: … hyperosmotic stresses, as well as…

Page 19, line 344: … causes oxidative stress…

Page 20, line 399: Thus, the differential expression of these genes could play…

Page 21, line 414: … WRKY TFs (WRKY24 and WRKY71)…

Page 23, line 507: … 40 Petri-dishes…

Page 23, lines 517-523: Here, it should be “absorbance reading was taken…”

Page 24, lines 532-538: Here, it should be “absorbance reading was taken…”

Author Response

Thanks again for your kind comments. Attachments below with our responses are shown in bold. Revised parts are marked up using the "Track Changes" function in the manuscript. 

1. The description of some results is very complex. It needs a considerable improvement to become easy to understand for readers. Language of the manuscript also needs to be improved, as it involves several grammatical, spelling, and punctuation mistakes.

According to your comments, the description of some results has been revised. For example, the “As shown in Figure 2, significant changes of activities of the four antioxidant enzymes (i.e., SOD, POD, CAT, and APX) in seeds were observed at different NaCl treatments with a significant increase with NaCl concentrations from 0 (CK) to 150 mmol/L, while a significant decrease with NaCl concentrations from 150 to 300 mmol/L”;  and the “As shown in Figure 3, significant changes in osmolytes contents (i.e., soluble sugar, protein, MDA, and Pro) in seeds were  observed at different NaCl treatments, with a significant increase for the contents of soluble sugar, MDA and Pro with the increased NaCl concentrations; while a significant decrease of the protein content was observed by the NaCl concentrations from 100 to 300 mmol/L.” (Page 3, lines 98-101; 107-111)

2. Page 2, line 59: “…while the DR of genes involved in cell wall structure”. This statement is not clear, please re-write it.

The sentence of “…while the DR of genes involved in cell wall structure” has been revised to “salt stress induced the UR of genes involved in cation transport and antioxidants, while induced the DR of genes involved in cell wall structure” in the manuscript. (Page 2, lines 57-58)

3. Page 2, line 74: It should be up regulated.

According to your comments, the “UR” has been revised to “up-regulated” in the manuscript. (Page 2, line 73)

4. Page, line 81: “while the adaptive mechanism to salt stress is still limited”. Please do not remove this statement, it should be included.

According to your comments, the sentence of “while the adaptive mechanism to salt stress is still limited” has been added in the manuscript. (Page 2, line 79)

5. Page 2, lines 81-85: “In order to reveal the adaptive…treatments were examined.” Please re-write this sentence to “In order to reveal the adaptive mechanism of A. venetum to salt stress in this study, we examined the changes into the rate of seed germination, activity of antioxidant enzymes, content of osmolytes, and expression level of genes in A. venetum at seed germination stage under different NaCl treatments.

The sentence of “In order to reveal the adaptive…treatments were examined” has been revised to “In order to reveal the adaptive mechanism of A. venetum to salt stress in this study, we examined the changes into the rate of seed germination, activity of antioxidant enzymes, content of osmolytes, and expression level of genes in A. venetum at seed germination stage under different NaCl treatments” in the manuscript. (Page 2, lines 79-83)

6. Page 2, lines 89-90: It should be “low NaCl concentration (50 mmol/L)…”

The describe of “low NaCl concentrations (50 mmol/L)…” has been revised to “low NaCl concentration (50 mmol/L)…”in the manuscript. (Page 2, line 87)

7. Page 3, lines 107-109: …significant changes in osmolyte contents (i.e., soluble sugar, protein, MDA, and Pro) was observed in seeds at different NaCl treatments, with a significant…

The sentence of “significant changes of contents of osmolytes (i.e., soluble 105 sugar, protein, MDA, and Pro) in seeds were also observed at different NaCl treatments 106 (0-300 mmol/L, p < 0.05), with a significant…” has been revised to “significant changes in osmolytes contents (i.e., soluble sugar, protein, MDA, and Pro) was observed in seeds at different NaCl treatments, with a significant …”in the manuscript. (Page 3, lines 107-110)

8. Page 3, line 112: mmol/L. While a significant decrease was observed by increasing the NaCl concentrations…

The sentence of “while a significant decrease with increased NaCl concentrations from 100 to 300 mmol/L for the protein content” has been revised to “while a significant decrease of the protein content was observed by  the NaCl concentrations from 100 to 300 mmol/L” in the manuscript. (Page 3, lines 110-111)

9. Page 4, line 120: In order to find far more genes…

The sentence of “In order to find our far more genes…” has been revised to “In order to find far more genes…” in the manuscript. (Page 4, line 149)

10. Page 5, line 146: …in Figure 4C. The distinct difference in…

The sentence of “in Figure 4C, The distinct difference in…” has been revised to “…in Figure 4C. The distinct difference in…” in the manuscript (Page 5, line 176)

11. Page 6, line 166: It should be “Specific Classification of DEGs and Validation of Expression Levels”

The title of “Specifically Classification of DEGs and Validation of Expression Levels” has been revised to “Specific Classification of DEGs and Validation of Expression Levels” in manuscript. (Page 6, line 198)

12. Page 13, line 241: … 8 selected genes were validated…

Thanks for your comments, the sentence of “The expression level of 8 select genes… ” has been revised to “The expression levels of 8 selected genes…” in the manuscript. (Page 11, line 273)

13. Page 19, line 312: … low NaCl concentration (50 mmol/L)…

The sentence of “…low NaCl concentrations (50 mmol/L) could…” has been revised to “…low NaCl concentration (50 mmol/L) could…” (Page 16, line 343)

14. Page 19, line 343: … hyperosmotic stresses, as well as…

The sentence of “…hyperionic and hyperosmotic stress, as well as …” has been revised to “…hyperionic and hyperosmotic stresses, as well as …” in the manuscript. (Page 16, line 369)

15. Page 19, line 344: … causes oxidative stress…

The sentence of “… cause oxidative stress…” has been revised to “… causes oxidative stress…” in the manuscript. (Page 16, line 370)

16. Page 20, line 399: Thus, the differential expression of these genes could play…

The sentence of “Thus, these genes differential expression could play…” has been revised to “Thus, the differential expression of these genes could play…” in the manuscript. (Page 17, line 424-425)

17. Page 21, line 414: … WRKY TFs (WRKY24 and WRKY71)…

The sentence of “…WRKYs (WRKY24 and WRKY71) …” has been revised to “…WRKY TFs (WRKY24 and WRKY71) …”in the manuscript. (Page 17, line 439)

18. Page 23, line 507: … 40 Petri-dishes…

The sentence of “… 40 Petri-dishs…” has been revised to “ 40 Petri-dishes…” in the manuscript. (Page 19, line 532)

19. Page 23, lines 517-523: Here, it should be “absorbance reading was taken…”

The sentence of “absorbance read was taken…” has been revised to “absorbance reading was taken…” in the manuscript. (Page 20, lines 544-550)

20. Page 24, lines 532-538: Here, it should be “absorbance reading was taken…”

The sentence of “absorbance read was taken…” has been revised to “absorbance reading was taken…” in the manuscript. (Page 20, line 557-564)

In addition, we have tried our best to read and checked the whole manuscript to avoid the grammatical, spelling, and punctuation mistakes.

If any question remains, please let us know.